# Understanding the dependence of mean precipitation on convective treatment and horizontal resolution in tropical aquachannel experiments

**Hyunju Jung**[1], **Peter Knippertz**[1], **Yvonne Ruckstuhl**[2], **Robert Redl**[2], **Tijana Janjic**[2,3], **and Corinna Hoose**[1]

[1]Institute of Meteorology and Climate Research (IMK), Department Troposphere Research,
Karlsruhe Institute of Technology (KIT), Karlsruhe, Germany
[2]Meteorological Institute, Ludwig Maximilian University of Munich, Munich, Germany
[3]Mathematical Institute for Machine Learning and Data Science, Katholische
Universität Eichstätt-Ingolstadt, Ingolstadt, Germany

**Correspondence:** Hyunju Jung (hyunju.jung@kit.edu)

**Abstract.** The Intertropical Convergence Zone (ITCZ) is a key circulation and precipitation feature in the tropics. There has been a large spread in the representation of the ITCZ in global weather and climate models for a long time, the reasons for which remain unclear. This paper presents a novel approach with which we disentangle different physical processes responsible for the changeable behavior of the ITCZ in numerical models. The diagnostic tool is based on a conceptual framework developed by Emanuel (2019) and allows for physically consistent estimates of convective mass flux and precipitation efficiency for simulations with explicit and parameterized convection. We apply our diagnostic tool to a set of tropical aquachannel experiments using the ICOsahedral Nonhydrostatic (ICON) model with horizontal grid spacings of 13 and 5 km and with various representations of deep and shallow convection. The channel length corresponds to the Earth's circumference and has rigid walls at 30° N/S. Zonally symmetric sea surface temperatures are prescribed.

All experiments simulate an ITCZ at the Equator coinciding with the ascending branch of the Hadley circulation and descending branches at 15° N/S with subtropical jets and easterly trade wind belts straddling the ITCZ. With explicit deep convection, however, rainfall in the ITCZ increases and the Hadley circulation becomes stronger. Increasing horizontal resolution substantially reduces the rainfall maximum in the ITCZ, while the strength of the Hadley circulation changes only marginally. Our diagnostic framework reveals that boundary-layer quasi-equilibrium (BLQE) is a key to physically understanding those differences. At 13 km, enhanced surface enthalpy fluxes with explicit deep convection are balanced by increased convective downdrafts. As precipitation efficiency is hardly affected, convective updrafts and rainfall increase. The surface enthalpy fluxes are mainly controlled by mean surface winds, closely linked to the Hadley circulation. These links also help understand rainfall differences between different resolutions. At 5 km, the wind–surface-fluxes–convection relation holds, but additionally explicit convection dries the mid-troposphere, which increases the import of air with lower moist static energy into the boundary layer, thereby enhancing surface fluxes. Overall, the different model configurations create little variations in precipitation efficiency and radiative cooling, the effects of which are compensated for by changes in dry stability. The results highlight the utility of our diagnostic tool to pinpoint processes important for rainfall differences between models, suggesting applicability for climate model intercomparison projects.

## 1 Introduction

Moist convection is of paramount importance in the tropics because it controls the distribution of water vapor, clouds and rainfall (Webster, 2020). Also, its importance lies in multi-

scale interactions with other processes, ranging from turbulence and microphysical processes via radiation and surface fluxes to large-scale circulations such as the Hadley cells that comprise the overturning circulation between 30° N and 30° S with an ascending branch at the Equator. One of the examples that illustrate the complexity of processes associated with moist convection is the so-called Intertropical Convergence Zone (ITCZ) (Schneider et al., 2014). Over oceans, the ITCZ is collocated with low-level convergence and upper-level divergence of the Hadley circulation accompanied by low-level easterly trade winds on the flanks (Johnson et al., 1999; Schwendike et al., 2014). Thermodynamic contrasts between the ocean and the air and surface winds modulate surface enthalpy fluxes, of which enhancement increases rainfall by transporting moisture and heat from the ocean into the air (Raymond et al., 2006; Paccini et al., 2021). Cumulonimbus clouds and clear-sky, moist columns in the tropics trap outgoing longwave radiation, and the moist columns increase the shortwave absorption, while the dry columns and shallow clouds in the subtropics enhance net longwave cooling compared to the tropics (Bony et al., 2015; Lau et al., 2020). An important parameter to characterize atmospheric behavior in the tropics is precipitation efficiency, the fraction of rain produced for a given amount of condensate. It has been shown that precipitation efficiency is linked to the ratio of cirrus to deep convective clouds (Stephens, 2005). The area fraction of these two cloud types modulates outgoing longwave radiation, which in turn controls the Earth's energy budget (Lindzen et al., 2001; Hartmann and Michelsen, 2002; Mauritsen and Stevens, 2015).

Climatologically, the location of the ITCZ is slightly shifted into the Northern Hemisphere (Webster, 2020). However, state-of-the-art general circulation models (GCMs) still struggle to accurately represent many characteristics of the ITCZ including the double-ITCZ problem leading to excessive rainfall in the Southern Hemisphere (Fiedler et al., 2020; Tian and Dong, 2020). Even in an idealized aquaplanet configuration, which avoids complexities associated with the land–sea distribution and orography, the spatial and temporal distributions of mean precipitation are sensitive to the type of numerical model (Stevens and Bony, 2013; Rajendran et al., 2013; Landu et al., 2014; Benedict et al., 2017), vertical and horizontal resolution (Li et al., 2011; Retsch et al., 2017, 2019), and representation of convection (Möbis and Stevens, 2012; Nolan et al., 2016; Retsch et al., 2019; Rios-Berrios et al., 2022).

Most of the current weather and climate models employ parameterizations for shallow and deep convection. The former plays an important role for the exchange between the boundary layer (BL) and the free troposphere, particularly in relatively dry areas (Schlemmer et al., 2017; Naumann et al., 2019; Sakradzija et al., 2020), and the latter is key for rainfall generation and vertical energy transport through latent heat release and mixing with ambient air (Emanuel, 1994; Bechtold, 2017; Webster, 2020). Explicitly representing convection on the model grid and thus avoiding convection parameterization is thought to be promising to reduce errors by permitting multi-scale interactions between convection and large-scale circulations (Randall, 2013; Palmer and Stevens, 2019; Tomassini, 2020), but it requires high model resolution. Given specific purposes and computational resources, a horizontal grid spacing of < 10 km can be selected to resolve deep convection (Weisman et al., 1997; Hong and Dudhia, 2012; Prein et al., 2015) with some extreme limit of 100 m (Kwon and Hong, 2017; Jeevanjee, 2017). Current global weather models use horizontal grid spacing of about 10 km with parameterized deep and shallow convection (Becker et al., 2021; Gehne et al., 2022). It is now feasible and affordable to conduct regional to global simulations with explicit deep convection (Satoh et al., 2017; Stevens et al., 2019; Schär et al., 2020; Wedi et al., 2020). These convection-permitting models show some promising results, particularly in the tropics where baroclinic instability is of little relevance for weather systems. Explicit convection captures spatial and temporal variability of tropical rainfall more realistically compared to parameterized convection (Stevens et al., 2020). Wind-induced surface exchange of heat and moisture is also improved, as shown for the tropical Atlantic Ocean by Paccini et al. (2021). Moreover, explicit deep convection performs better in terms of convectively coupled equatorial waves (Judt and Rios-Berrios, 2021) and gravity wave momentum fluxes, which are often triggered by convection in the tropics and subtropics (Stephan et al., 2019).

Despite these many improvements, convection-permitting models do not always guarantee alleviating the long-standing ITCZ problem (Zhou et al., 2022). Furthermore, models with explicit deep convection do not outperform those with parameterizations in every aspect. Parameterized deep convection is in better agreement with observations than explicit deep convection in terms of mean rainfall distribution (Wedi et al., 2020). Furthermore, Becker et al. (2021) demonstrated that their new convection parameterization scheme, which improves the coupling of convection to mesoscale dynamics, outperformed explicit deep convection in terms of both mean and intensity of rainfall over tropical Africa. Jung and Knippertz (2023) showed that the representation of equatorial waves does not deviate much between explicit and parameterized deep convection when using a global forecast model. These results indicate that resolving (deep) convection does not automatically improve the multi-scale interactions in the atmosphere and does not necessarily reduce the bias in tropical rainfall. In fact, it is crucial to accurately represent physical processes and links between them.

A general problem in this context is that it is far from trivial to disentangle the reasons for the difference in performance when switching from parameterized to explicit deep convection, which often includes changes in horizontal resolution, since convection couples and interacts with so many physical processes. To tackle this problem, we here propose an innovative diagnostic tool based on a conceptual frame-

work developed by Emanuel (2019). This framework is built around boundary-layer quasi-equilibrium (BLQE), the weak temperature gradient approximation, and mass and energy conservation. BLQE describes a balance of moist entropy in the subcloud layer. The balance is achieved between surface enthalpy fluxes, which transport warm, moist air into the subcloud layer, and convective downdrafts and environmental subsidence, which transport cool, dry air from the free troposphere into the subcloud layer (Emanuel et al., 1994; Raymond, 1995). The weak temperature gradient approximation neglects horizontal temperature advection, implying a balance between diabatic heating and vertical advection (Sobel et al., 2001). Emanuel's (2019) framework considers processes on a timescale longer than that associated with the redistribution of energy by internal gravity waves. A key parameter of the conceptual model is precipitation efficiency that summarizes the collective effects of turbulent and microphysical processes. Despite its relative simplicity, the framework is able to explain fundamental characteristics of the tropical atmosphere such as the exponential relationship between rainfall and column relative humidity (Bretherton et al., 2004), convective self-aggregation (Bretherton et al., 2005), and the horizontal structures of the Walker and Hadley circulations. We refer to Emanuel (2019) for further demonstrations of atmospheric phenomena in his framework.

The goal of our study is to disentangle the physical processes contributing to differences in the ITCZ when modifying the model configuration. The modifications include changes in horizontal resolution and the representation of deep and shallow convection. To avoid the complexity associated with continents, orography, zonal asymmetries and influences of the extratropics on the tropical conditions, we conducted a set of tropical aquachannel experiments with time-constant, Equator-symmetric sea surface temperatures (SSTs). The simulations are as realistic as possible by including a latitude-dependent Coriolis parameter and a diurnal cycle in solar irradiance. Section 2 explains further details of the model and experimental design. Section 3 describes the large-scale behavior of the aquachannel experiments and differences due to model configuration. Section 4 presents our diagnostic approach based on Emanuel (2019). Results from applying the new approach to the tropical aquachannel simulations are shown and discussed in Sect. 5. Conclusions are given in Sect. 6.

## 2 Aquachannel experiments

### 2.1 Model

We use version 2.6.3 of the ICOsahedral Nonhydrostatic (ICON) model (Zängl et al., 2015) in the numerical weather prediction (NWP) configuration. The model solves the fully compressible nonhydrostatic atmospheric equations of motion on an icosahedral–triangular Arakawa-C grid. Radia-

tion is computed using the Rapid Radiative Transfer Model (RRTM) (Mlawer et al., 1997). A single-moment microphysical scheme is used to predict cloud water, rain water, cloud ice and snow (Seifert, 2008). A turbulent kinetic energy scheme is used for the representation of turbulent mixing and surface-to-atmosphere transfer (Raschendorfer, 2001; Mellor and Yamada, 1982). Our model configuration closely follows the operational setup, including a full non-linear Coriolis parameter, but some aspects are different for the specific purpose of our study. The surface of the entire model domain is covered by water (aquaplanet or aquachannel simulation) to exclude the complexity associated with topography, and the diurnal cycle has fixed equinoctial insolation over the whole simulation period. Zonally symmetric SSTs are prescribed with a maximum of 27 °C at the Equator dropping to approximately 5 °C at 60° N/S. This SST distribution has been used in other studies and is called the "Qobs" profile (Neale and Hoskins, 2000). There is no feedback of the atmosphere on the ocean and the underlying water surface, effectively making the ocean an indefinite energy source.

### 2.2 Simulation setup

To spin up our aquachannel simulations, we adapt the modeling practice used in Bretherton and Khairoutdinov (2015). First, we conduct a global aquaplanet simulation with a horizontal grid spacing of 40 km and a time step of 300 s. The initialization of the 40 km aquaplanet run follows the Qobs case of Neale and Hoskins (2000). The number of vertical levels is 90 with the model top at 75 km. Deep and shallow convection are parameterized using a bulk mass flux scheme (Bechtold et al., 2008; Tiedtke, 1989). The 40 km global aquaplanet experiment is run for 120 simulation days (dashed black line in Fig. 1), after which the grid spacing is reduced to 26 km with a time step of 225 s and the simulation is continued for another 90 d. After that, the model domain is restricted to a channel geometry between 30° N and 30° S, and the horizontal grid spacing is reduced to 13 km with a time step of 112.5 s (dotted black line in Fig. 1). The domain encloses the entire globe and forms a closed ring in the zonal direction. Closed walls are introduced at the latitudinal boundaries where virtual potential temperature, water vapor mixing ratio, air density, and zonal and vertical winds are prescribed by zonally and temporally averaging them at 30° N and 30° S from the 26 km aquaplanet simulation. The prescribed variables at the closed walls are time-invariant and zonally constant but vertically variant. Except for the aforementioned quantities, all other variables are set to zero at the walls. The setup for the aquachannel run is identical to the aquaplanet runs except for the simulation geometry and the horizontal resolution. The coarser aquaplanet simulations thus serve to obtain the boundary conditions and to spin up the aquachannel run with reduced computational cost. The total simulation period of the 13 km aquachannel run is 102 d, consisting of spin-up at the beginning of 62 d and the analysis period

of 40 d. Finally, the grid spacing is reduced to 5 km at day 314 with a time step of 45 s (dotted pink line in Fig. 1). The boundary conditions of the high-resolution run are identical to the 13 km run. The analysis period of the final run is 40 d, starting at day 317 (solid pink line in Fig. 1), after a 3 d spin-up. The output time step of the 13 and 5 km aquachannel runs is hourly.

To illustrate the modeling approach, Fig. 1 depicts the evolution of the probability density distribution of precipitable water (PW) in the equatorial belt (20° N/S) over the entire run time from day 0 to 357. In the beginning of the 40 km aquaplanet simulation, PW is distributed narrowly around 40 kg m$^{-2}$, but by day 50 the distribution has widened with a broad dry maximum around 25 kg m$^{-2}$ and a narrower secondary maximum near 55 kg m$^{-2}$. After that, the bimodal shape remains stable, even when the grid spacing is reduced from 40 to 26 km on day 120. The moist maximum corresponds to the actual ITCZ region, while the dry maximum represents the large area of subsidence in the cooler outer tropics with relatively few intermediate values of PW in between. Such a rapid evolution into a stable bimodal structure was seen in other aquaplanet simulations with the Coriolis force (e.g., Arnold and Randall, 2015; Khairoutdinov and Emanuel, 2018), as the large-scale circulation redistributes moisture from the relatively homogeneous initial conditions.

When the 13 km aquachannel experiment begins on day 210, a considerable change can be observed. The range of PW slowly decreases due to a reduction of the moist columns and an increase in the frequency of dry areas, despite little change in magnitude. This drift slows down but still continues into the investigation period after day 274, suggesting that a full equilibrium has not been reached yet. Towards the end of the coarse-resolution aquachannel simulation around day 314, there are some indications of a bimodal distribution again yet much closer to each other than in the global simulation before day 210. The reason for this behavior lies in the prescribed properties at the closed walls. While in the global configuration the Hadley cells span over 30° N/S, in the aquachannel configuration the model creates its own limited Hadley circulation away from the walls with subsidence around 15° N/S (described in Sect. 3.2). The narrower overturning circulation reduces the amount of moisture converging into the ITCZ (not shown). It is also conceivable that the suppression of exchange with the higher latitudes reduces moisture uptake through surface fluxes triggered by dry intrusions from the midlatitudes (Bretherton and Khairoutdinov, 2015).

Throughout the entire period of the 5 km aquachannel run (day 314–357), the bimodal distribution persists with the frequency density of PW confined between 20 and 50 kg m$^{-2}$. The remarkably smooth transition from the 13 to 5 km runs indicates that a change in horizontal resolution creates almost no distortion of the fields, which is also observed when reducing the grid spacing from 40 to 26 km (dashed black line in Fig. 1). This allows us to have relatively few days of spin-

up for the high-resolution run. In summary, the PW evolution over the entire simulation period exhibits smooth transitions not only from the coarse to high resolutions but also from the aquaplanet and aquachannel geometries.

Our original intention to prescribe zero meridional wind and constant zonal and vertical winds from the 26 km aquaplanet experiment at the rigid walls was to simulate a Hadley circulation with descending branches near 30° N/S as in the global runs, but the model develops its own Hadley circulation rather than connecting its dynamical fields with the boundaries. We suspect that a possible reason is the suppression of eddy transport between the tropics and extratropics at the boundaries, forcing the model to develop its own subtropical jets internally. Ultimately, this also leads to distortions in the fields of cloud, radiation and surface fluxes in the outer tropics. We presume that a wider channel, a two-way nested channel within a global domain or an aquaplanet would simulate jets at a more realistic location and may affect many aspects, particularly associated with tropical–extratropical interactions. However, the channel geometry suppresses these interactions and thus reduces complexity. Furthermore, the advantage of having jets at a more realistic location does not outweigh the merit of our configuration that is still able to reproduce a complex structure of dynamics and thermodynamics of the tropical atmosphere with affordable computational resources. To give as little weight as possible to the artifacts from the channel approach, we restrict our analysis to an equatorial belt between 20° N and 20° S (corresponds to the area used in Fig. 1). We are confident that our analysis for this area can give useful insights into how convective treatment and model resolution affect ITCZ processes, at least in a qualitative sense.

We experimentally modify the representation of deep and shallow convection in the aquachannel configuration. At 13 km, the different treatments of deep and shallow convection are described in the following way (see Table 1): (a) an experiment named P13 where the deterministic deep and shallow convection schemes are turned on (already shown in the context of Fig. 1), (b) E13 where both deep and shallow convection schemes are turned off, (c) S13 where only the deep convection scheme is turned off and (d) SS13 where the standard deterministic shallow convection scheme (Bechtold et al., 2008; Tiedtke, 1989) is replaced by a stochastic scheme (Sakradzija et al., 2015, 2020). In the stochastic shallow convection scheme, the shallow-cloud ensemble is represented based on the theory of Craig and Cohen (2006). The number of new clouds is set using a Poisson distribution and the lifetime average mass flux using a Weibull distribution. In the stochastic scheme, there are two constraints: the mass flux closure of the deterministic scheme to constrain the ensemble average mass flux and the surface Bowen ratio to control the average mass flux per cloud (Sakradzija and Hohenegger, 2017). Note that the four experiments with the horizontal grid spacing of 13 km share the same aquaplanet runs as spin-up and that we modify the convective treatment when the chan-

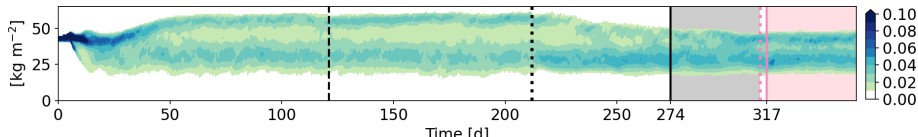

**Figure 1.** Evolution of the frequency density distribution of precipitable water in the equatorial belt between 20° N and 20° S over a successive set of aquaplanet and aquachannel simulations including P13 and P5 in Table 1. The dashed black line indicates when the horizontal grid spacing is reduced from 40 to 26 km, the dotted black line from 26 to 13 km and the dotted pink line from 13 to 13 km. The solid lines indicate when the analysis periods of 40 d each begin. Colored shading in the background indicates the analysis period.

**Table 1.** Horizontal grid spacing and treatment of deep and shallow convection schemes for each experiment.

| Exp. name | $\Delta x$ [km] | Deep conv. | Shallow conv. |
|-----------|-----------------|------------|----------------|
| P13 | 13 | On | On (deterministic) |
| E13 | 13 | Off | Off |
| S13 | 13 | Off | On (deterministic) |
| SS13 | 13 | Off | On (stochastic) |
| P5 | 5 | On | On (deterministic) |
| E5 | 5 | Off | Off |

nel geometry is introduced (dotted black line at day 210 in Fig. 1). Other than the different representations of convection, the setups remain identical among the coarse-resolution aquachannel experiments.

At day 314, the horizontal grid spacing is further reduced to 5 km, creating two high-resolution runs (Table 1): P5 with the deterministic deep and shallow convection schemes and E5 with explicit deep and shallow convection. Both high-resolution simulations are initialized with P13 at day 314, but spin-up is 2 d longer for E5 than P5, the analysis period of which begins at day 317 (solid pink bar in Fig. 1). The boundary conditions for the high-resolution runs are identical to the coarse-resolution ones.

In the following sections, we analyze the last 40 d of each aquachannel experiment, e.g., after spin-up (day 274–314 for the coarse-resolution runs). To compare the experiments with different horizontal resolutions, model grids are coarsened on a 0.2° latitude–longitude grid, using a conservative remapping.

## 3 Overview of aquachannel experiments

### 3.1 Precipitation

The latitudinal distributions of zonally and time-averaged precipitation are shown in Fig. 2a. At 13 km, all experiments show a distinct ITCZ with high mean precipitation concentrated between 5° N and 5° S, where the SST maximum is prescribed. Explicit deep convection (E13, S13, and SS13) yields greater mean precipitation in the ITCZ than parameterized deep convection (P13) by about 35 %. Between 5° N

and 5° S the time and zonally averaged precipitation is 7.28, 9.76, 9.76 and 9.64 mm d$^{-1}$ for P13, E13, S13 and SS13, respectively. P13 also produces a narrower, more pointy rainfall distribution. The treatment of shallow convection does not appear to have a large influence on the mean ITCZ structure. The 5 km experiments are generally drier and show indications of a broader – even double – ITCZ. As observed for the coarse-resolution runs (13 km) but to a lesser extent, E5 (7.5 mm d$^{-1}$) has higher mean rainfall in the ITCZ than P5 (6.72 mm d$^{-1}$). The shape of mean rainfall is fairly symmetric in E5, yet rainfall clearly favors the Northern Hemisphere in P5. These runs are initialized with P13, which produces a disturbance similar to the Madden–Julian Oscillation in the last 20 d with higher rainfall in the Northern Hemisphere than in the Southern Hemisphere (not shown). This initial asymmetry appears to have a long-lasting effect in both runs (see ITCZ broadening around 7.5° N) but particularly in P5. We plan to conduct a more detailed analysis on internal variability in the future.

Outside the ITCZ, the overall rainfall amount and the differences between the experiments are relatively small. At around 10° N/S rainfall decreases to about 1 mm d$^{-1}$, and beyond this it slightly increases again with latitude. This pattern of rainfall in the outer tropics is also observed in other aquaplanet, aquachannel and aquapatch simulations (e.g., Nolan et al., 2016; Rios-Berrios et al., 2022). It is due to rainfall embedded in filaments of high PW being sheared off from the ITCZ into the outer tropics (see the "Video supplement").

The rainfall intensity distribution further underlines the substantial sensitivities to convective treatment and resolution (Fig. 2b). Comparing the results among the 13 km runs, light and moderate rains ($< 30$ mm d$^{-1}$) occur more frequently in P13 than the others, which produce extreme rainfall rates of 200 mm d$^{-1}$ and more, leading to the overall larger precipitation in the ITCZ. Going from 13 to 5 km, the frequency of intense rainfall decreases; E5 shows a discernible reduction compared to E13, while P5 shows a small decrease compared to P13. This leads to a smaller difference in rainfall intensity between P5 and E5 compared to the coarse-resolution runs. This resolution dependency differs from Becker et al. (2021), who showed that rainfall intensity over tropical Africa is not dependent on resolution but on convective treatment (see their Fig. 3).

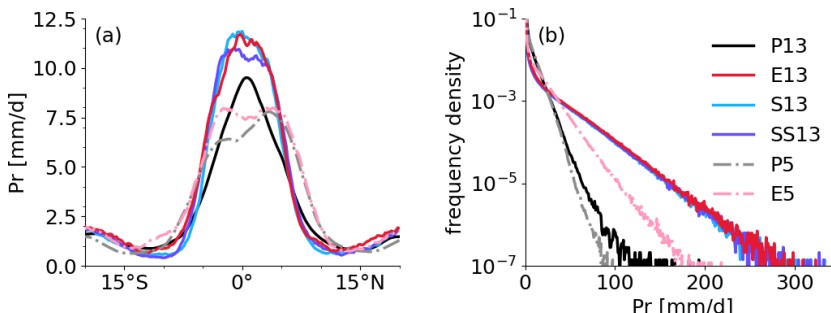

**Figure 2.** Distributions of **(a)** time and zonal mean of precipitation rate (Pr) against latitude and **(b)** frequency of different precipitation intensities between $20°$ N and $20°$ S. For the intensity distribution, daily precipitation is grouped with a bin size of $1\,\mathrm{mm\,d^{-1}}$. Note the logarithmic vertical axis in panel **(b)**.

Correspondingly, the "Video supplement" depicts that large-scale systems of precipitation with weak intensity are formed with parameterized deep convection (P13 and P5), whereas intense, localized storms are formed with explicit deep convection (E13, E5, S13 and SS13). The much higher intensities also lead to a more wiggly zonal average, as evident in Fig. 2a. To initiate deep convection explicitly, the model needs to develop instability on a grid box scale. The larger the grid box (or the coarser the grid resolution), the more instability can be accumulated over time, which in turn produces more intense rainfall (Weisman et al., 1997) and occasionally intense grid point storms (Giorgi, 1991; Scinocca and McFarlane, 2004). Meanwhile, a convection parameterization scheme triggers convection by perturbing temperature and humidity profiles at low levels, which allows P13 and P5 to produce light and moderate rain more frequently. This difference in rainfall intensity between parameterized versus explicit deep convection was also observed in realistic simulations (Pante and Knippertz, 2019; Judt and Rios-Berrios, 2021).

### 3.2 Dynamical structure

Figure 3 shows a cross section of the time mean meridional-height mass stream function and zonal wind. The meridional-height mass stream function is calculated by integrating the meridional wind from the surface level to a certain altitude. Volumetric flux is conserved along a line of a constant meridional-height mass stream function. P13 features a largely Equator-symmetric troposphere-deep Hadley circulation with low-level convergence and corresponding upper-level divergence in the ITCZ (Fig. 3a). The remaining small asymmetries, which occur despite the symmetric nature of our simulation setup, are a further indication that the simulations may not have fully reached equilibrium or that there can be spontaneous symmetry breaking through internal variability. The descending branches occur around $15°$ N/S, which is narrower than the climatological Hadley circulation in the real atmosphere (Webster, 2020) and in global aquaplanet simulations (not shown). The narrower Hadley circulation

in the aquachannel experiments is because the exchange between the tropics and extratropics is suppressed at the closed walls of the tropical channel (discussed in Sect. 2.2). Strong westerly upper-tropospheric jets occur at the outer edges of these narrow Hadley cells, reaching an average speed of $30\,\mathrm{m\,s^{-1}}$. These in principle resemble the subtropical jets of the real atmosphere but shifted closer to the Equator and weaker. The low-level easterly trade wind belt starts at about $14°$ N/S and reaches about $2\,\mathrm{km}$, above which westerlies dominated. This creates a considerable westerly shear for the ITCZ convection (see the "Video supplement").

The other experiments (Fig. 3b–f) generally produce similar large-scale dynamical structures to P13. However, the strength of the overturning circulation and accompanying jets depends on convective treatment and on horizontal resolution. At $13\,\mathrm{km}$, explicit deep convection increases the maximum value of the mass stream function to $1.84 \times 10^{11}$, $1.81 \times 10^{11}$ and $1.82 \times 10^{11}\,\mathrm{kg\,s^{-1}}$ for E13, S13 and SS13, respectively, compared to P13 ($1.3 \times 10^{11}\,\mathrm{kg\,s^{-1}}$). This tendency is also present in the high-resolution runs, leading to maximum values of the mass stream function of $1.36 \times 10^{11}$ and $1.67 \times 10^{11}\,\mathrm{kg\,s^{-1}}$ for P5 and E5, respectively. The magnitude of the simulated circulation is in agreement with other aquaplanet studies (Medeiros et al., 2016; Rios-Berrios et al., 2020), but P13 is at the lower end of the range found in these studies. The stronger large-scale circulation with explicit deep convection at $13\,\mathrm{km}$ is accompanied by stronger trade winds, with an increase in surface horizontal wind speed to about $4\,\mathrm{m\,s^{-1}}$. This largely agrees with Paccini et al. (2021), who showed an increase in surface winds from a parameterized low-resolution run to an explicit high-resolution run using the ICON-NWP in a realistic setup. In contrast, the trade wind speed is not much influenced by the convective treatment at $5\,\mathrm{km}$, although the change in the large-scale circulation is considerable. Additionally, low-level zonal winds exhibits asymmetry, possibly due to the long-term memory that is also evident in the ITCZ. The asymmetry in surface winds at $5\,\mathrm{km}$ is shown clearly in Sect. 5.1.1.

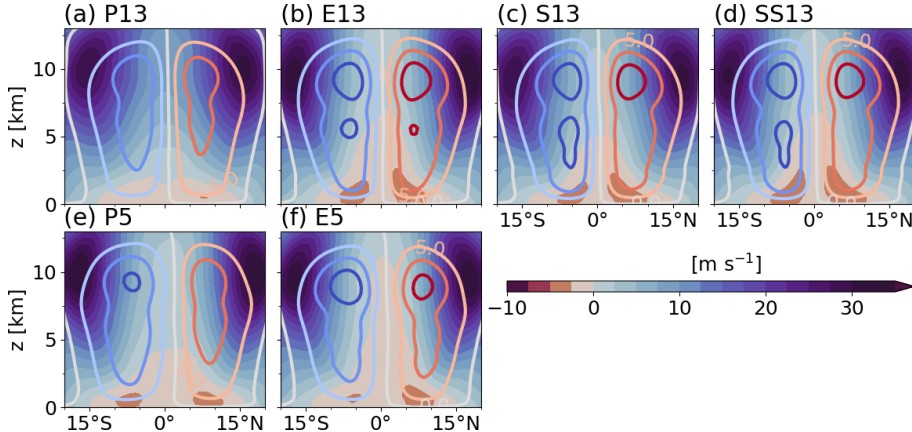

**Figure 3.** Time and zonal mean of zonal wind [m s$^{-1}$] (shading) and meridional-height mass stream function [$10^{10}$ kg s$^{-1}$] (contour lines) for the analysis period in experiment **(a)** P13, **(b)** E13, **(c)** S13, **(d)** SS13, **(e)** P5 and **(f)** E5. The interval for the colored contours is $5 \times 10^{10}$ kg s$^{-1}$.

One interesting aspect is that the runs with explicit deep convection (E13, S13, SS13 and E5) exhibit equatorial easterlies in the mid-troposphere up to 5–7 km, while P13 and P5 exhibit westerlies there. Possibly, the explicit deep convection produces more upward convective momentum transport. This mechanism may also weaken the westerlies in the upper troposphere, leading to an overall much enhanced horizontal wind shear towards the outer tropics. The vertical shear in contrast is reduced in the ITCZ with potential consequences for intense, short-lived rainfall (see the "Video supplement") because the sufficient shear is needed to generate long-lived organized systems (Wu and Moncrieff, 1996). There are also some subtle differences in the strength and depth of the trade wind layer, supporting the idea that vertical momentum transport may play a role.

The coarse-resolution runs with explicit deep convection (E13, S13 and SS13) generate a bimodal structure in the mass stream function, indicating a secondary shallow circulation that diverges polewards at around 7.5 km (Fig. 3b–d). Such a shallow meridional circulation is observed in the eastern Pacific, but the flow diverges at lower altitudes than in our experiments (Zhang et al., 2004).

## 4 ITCZ diagnostic tool

This section presents our diagnostic tool based on Emanuel's (2019) framework, which we apply in Sect. 5 to output from the aquachannel experiments to further explore the discussed differences mainly in rainfall. Amongst the three equations in the original framework, we only use the formulation of convective updraft mass flux, which can be directly related to precipitation. We refer to Emanuel (2019) for the complete derivation of the conceptual framework.

The framework assumes boundary-layer quasi-equilibrium (BLQE; Raymond, 1995), the weak temperature gradient approximation (Sobel et al., 2001) and energy and mass conservation, and it neglects horizontal advection of moist static energy in the BL. Using these assumptions, the framework formulates convective updraft mass flux $M_u$ as follows:

$$M_u = \frac{1}{1 - \epsilon_p}\left(\frac{F_h}{h_b - h_m} - \frac{\dot{Q}}{S}\right), \tag{1}$$

where $M_u$ is the convective upward mass flux of water vapor in kg m$^{-2}$ s$^{-1}$; $\epsilon_p$ is the precipitation efficiency; $F_h$ is the surface enthalpy flux; $\dot{Q}$ is the radiative cooling; $h_b$ and $h_m$ are the moist static energy of the BL and the free troposphere, respectively; and $S$ is the dry stability. Moist static energy is defined as $h = \phi + c_p T + L_v q_v - L_f q_i$, with $\phi$ being geopotential, $c_p$ the specific heat at constant pressure, $T$ the temperature, $L_v$ the latent heat of vaporization, $L_f$ the latent heat of freezing, $q_v$ the specific humidity and $q_i$ the specific ice content. Equation (1) demonstrates that the convective updraft mass flux increases with increasing surface enthalpy flux, with decreasing the vertical difference in moist static energy, with decreasing radiative cooling and with increasing dry static stability. The complete set of the conceptual framework for $M_u$ (Emanuel, 2019) is included in Appendix A.

One important parameter of the framework is $\epsilon_p$, which represents the fraction of all the condensate that reaches the ground as precipitation. Microphysical processes are not treated explicitly but formulated through one constant value of $\epsilon_p$. Also, $\epsilon_p$ is used to parameterize convective downdraft mass flux $M_d$ as a function of $M_u$ in the following way: $M_d = (1 - \epsilon_p)M_u$. For $\epsilon_p = 1$, all condensate precipitates, such that there is no evaporation and thus no downdraft mass flux. For $\epsilon_p = 0$, all condensate eventually evaporates again, such that downdraft and updraft mass fluxes balance.

For simplicity, Emanuel (2019) assumed that the average of the tropospheric $\dot{Q}$ can be approximated with the radiative cooling at the top of the BL in order to couple the bud-

get of $h$ and the large-scale thermodynamic balance (see details in Appendix A). Raymond et al. (2015) suggested using a lower tropospheric quasi-equilibrium instead of the entire tropospheric adjustment because when $h_b$ increases from its equilibrium, the lower troposphere responds to the deviation on a convective timescale. We thus average $h$ in the lower troposphere between 0.5 and 5 km to obtain a typical value $h_m$, and the same layer is considered for $\dot{Q}$, which is an averaged quantity, and $S$, which represents a slope of dry static energy. For computing $h_b$, we average $h$ from the lowest atmospheric level of 10 m to an approximate BL top of 500 m. We tested alternatives for the BL in the range from 0.4 to 1.5 km and for the troposphere from 4 to 9.5 km, and we found the main findings to be rather insensitive to the exact choice of altitudes (not shown).

To relate this conceptual framework to our physical output, we need to find a relation between the modeled precipitation (either explicit or parameterized) and convective mass flux. We assume that precipitation rate Pr is directly proportional to $M_u$ and $\epsilon_p$:

$$\mathrm{Pr} = \epsilon_p M_u \langle q_v \rangle, \tag{2}$$

with $\langle q_v \rangle$ being the column specific humidity. The notation $\langle X \rangle$ indicates the mass-weighted column mean quantity, $\int \rho X \, dz / \int \rho \, dz$. Precipitation can be related to the water vapor concentration at the subcloud layer or the average specific humidity of the subcloud layer rather than $\langle q_v \rangle$. We tested different choices of the thermodynamic variable in Eq. (2), but it does not influence our main results – only their scale in magnitude.CE1

Using Eqs. (1) and (2), we have two unknowns, $M_u$ and $\epsilon_p$, because the other quantities can be readily obtained from the model output and we can solve for them. In principle $M_u$ could be calculated from $w$ for each simulation, but vertical motions of explicit and parameterized convection contain different processes. Parameterized convection assumes a profile of $w$ through convective adjustment (Tiedtke, 1989), whereas explicit $w$ is computed from the dynamical core. Therefore, comparing these two motions directly from the model output is a comparison of apples and oranges. The same principle applies to $\epsilon_p$, which is related to $M_u$. In our diagnostic tool, $M_u$ and $\epsilon_p$ are not obtained directly from vertical motion but indirectly using other consistent quantities. In other words, $F_h$, $h_b - h_m$, $\dot{Q}$, $S$, Pr and $\langle q_v \rangle$ are fed into the two independent Eqs. (1 and 2) to estimate $M_u$ and $\epsilon_p$. In this way, the estimates are physically consistent across the experiments with different convective treatments.

# 5 Application

Section 3 showed substantial differences in the mean state, mainly due to the horizontal resolution and the deep convective treatment. Section 4 presented a diagnostic tool to compare these differences in a fair manner. Here we apply the

diagnostic tool to averaged fields over the analysis period of 40 simulation days, with a particular focus on mean rainfall. Given the zonal symmetry of our tropical channel, we mostly consider zonal means. In the following, we discuss the different aspects of the conceptual model after another: surface enthalpy fluxes (Sect. 5.1), vertical structure of moist static energy (Sect. 5.2), radiative cooling (Sect. 5.3), dry stability (Sect. 5.4), precipitation efficiency and convective mass flux (Sect. 5.5), and finally meridional advection in the BL (Sect. 5.6).

## 5.1 Surface enthalpy fluxes

The time and zonal mean of surface enthalpy fluxes is shown in Fig. 4a. P13 has $F_h$ maxima in the trades and a local minimum in the ITCZ (black line in Fig. 4a), similar to the situation over real-world tropical oceans but confined to a narrower latitudinal stretch. The coarse-resolution experiments with explicit deep convection (E13, S13 and SS13) share similar latitudinal distributions, but $F_h$ increases compared to P13, in particular between 10° N and 10° S (by 20 %–25 %). Note that shallow convection is represented by the deterministic shallow convection scheme for S13, by the stochastic shallow convection scheme for SS13 and explicitly for E13 (Sect. 2.2). At 13 km, therefore, the main difference in $F_h$ is due to the treatment of deep convection rather than shallow convection. The dependence of $F_h$ on convective treatment is consistent with that of the Hadley circulation and thus surface winds described in Sect. 3.2. The high-resolution experiments exhibit a similar $F_h$ distribution and dependence on convective treatment to the coarse-resolution ones. However, $F_h$ is enhanced less strongly between 10° N and 10° S with explicit convection and has the deeper local minima in the ITCZ, leading to an increase from P5 to E5 (by 11 %). The resolution dependence is more complex. From P13 to P5, $F_h$ is reduced in the ITCZ but enhanced in the trade wind zone, leading to an overall small increase by 2 %. From E13 to E5, $F_h$ is systematically reduced by 11 %. The difference between the runs remains smaller beyond 15° N/S than in the inner tropics. To investigate what controls these differences in surface enthalpy fluxes, we conduct a more detailed analysis. We decompose surface fluxes into their contributing factors (Sect. 5.1.1) and examine their statistical distribution (Sect. 5.1.2).

### 5.1.1 Decomposition of surface fluxes

In a standard air–sea bulk formula, surface enthalpy fluxes can be written as

$$F_h = \rho_s L_v c_E \overline{U}_h \Delta q + \rho_s c_p c_H \overline{U}_h \Delta T, \tag{3}$$

where $\rho_s$ is the air density at the lowest model level, $L_v$ is the latent heat of vaporization, $c_p$ is the specific heat at constant pressure, $c_E$ and $c_H$ are the surface exchange coefficients for latent and sensible heat, respectively, $\overline{U}_h$ is the sur-

face horizontal wind speed, and $\Delta q$ and $\Delta T$ are the air–sea moisture and temperature contrasts. For our analysis, we define $\Delta q = q_*(\text{SST}) - q_v(z_1)$ and $\Delta T = \text{SST} - T(z_1)$, where $q_*(\text{SST})$ is the saturated specific humidity for a given SST and $z_1$ indicates the lowest model level of the atmosphere, which equals to 10 m in our case. Here we begin with partitioning $F_h$ into surface sensible and latent heat fluxes to examine the importance of thermodynamic variables, i.e., $\Delta q$ and $\Delta T$ as well as $\overline{U}_h$ for mean $F_h$.

Figure 5a–c shows zonally and time-averaged values of the individual terms of Eq. (3). Figure 5a reveals that $\overline{U}_h$ mirrors the patterns in $F_h$ (Fig. 4a) with maxima in the trade winds and with minima at the Equator and in the area of the subsiding branches of the Hadley cells (Fig. 3). Winds then increase again further away from the Equator. P13 shows considerably weaker $\overline{U}_h$ by about $1\,\text{m}\,\text{s}^{-1}$ than the other coarse-resolution runs (E13, S13, and SS13) out to about $10°$ from the Equator. The treatment of shallow convection (S13 and SS13) appears to have a rather small influence on $\overline{U}_h$. In P5 and E5, the pattern of $\overline{U}_h$ exhibits asymmetry with stronger trade winds in the Northern Hemisphere and a deeper local minimum as evident in $F_h$ (Fig. 4a). The difference in $\overline{U}_h$ between these two runs is arguably small, albeit with stronger Hadley circulation in E5 than in P5 (Fig. 3e and f). This indicates that the surface wind speed is less sensitive to convective treatment at 5 km. Meanwhile, the differences in $\overline{U}_h$ due to resolution, e.g., between P13 and P5 and between E13 and E5, reflect those in surface enthalpy fluxes (Fig. 4a).

The moisture contrast shows a much smoother latitudinal distribution and considerable contrasts between all simulations, mainly due to convective treatment (Fig. 5b). In P13 and P5, $\Delta q$ is almost constant around $6.60\,\text{g}\,\text{kg}^{-1}$ within $15°$ N/S and then sharply falls off towards higher latitudes as $q_*$ quickly drops at these latitudes. The explicit treatment of deep convection (E13, S13, SS13 and E5) appears to allow for more vigorous downdrafts injecting dry air from the mid-troposphere into the BL. In contrast to other fields discussed so far, the treatment of shallow convection also plays a significant role. S13, which uses the same shallow convection scheme as P13 but no parameterization of deep convection, shows only slightly enhanced $\Delta q$, particularly in the trade wind zone, where shallow mixing is important. The change to the stochastic treatment (SS13) from the deterministic treatment (S13) has little effect in the moist ITCZ area but further enhances $\Delta q$ in the trades, eventually lining up with E13 at around $10°$ N/S. In E5 $\Delta q$ closely follows that in E13, but there is a discernible decrease by around $0.5\,\text{g}\,\text{kg}^{-1}$ in the trade wind zone in the Northern Hemisphere.

Figure 5c shows that the latitudinal structure of $\Delta T$ is complex in response to different convective treatments and resolutions. All simulations have a local maximum at the Equator, probably related to cool downdrafts from convection, but some have prominent maxima near the subsiding branches of the Hadley cells before all runs show a drop-off towards higher latitudes. E13 shows the overall smallest $\Delta T$, possibly because it produces deeper convective downdrafts, leading to more adiabatic warming during the descent. E5 closely follows this structure with a marginal increase in $\Delta T$. The two coarse-resolution simulations with parameterized shallow convection (S13 and SS13) largely agree with E13 near the Equator but show considerably larger $\Delta T$ in the outer tropics, in particular S13. The reasons for this are not entirely clear. P13 has relatively high $\Delta T$ at the Equator compared to the other coarse-resolution runs and intermediate values in the outer tropics. Finally, P5 has the highest $\Delta T$ at the Equator with a maximum of 2.3 K, meaning that the air near the surface is much colder in P5 than in the other runs. P5 shows the lowest rainfall intensity (Fig. 2b) and thus probably a higher fraction of subcloud evaporation, which cools and moistens the BL, leading to considerably high $\Delta T$ (Fig. 5c) as well as low $\Delta q$ (Fig. 5b). However, these differences in $\Delta T$ have little impact on the surface enthalpy fluxes, since the surface sensible heat flux contributes only about 10 %. (In the latitudinal belt of $20°$ N/S the time and domain mean of the latent heat flux accounts for $95.5\text{–}113.6\,\text{W}\,\text{m}^{-2}$, while the surface sensible heat flux is $9.2\text{–}11.1\,\text{W}\,\text{m}^{-2}$.)

Surface enthalpy fluxes can be modulated by mean winds and thermodynamics, as well as local perturbations of those components. To quantify this, the time and zonal mean of surface enthalpy fluxes (Fig. 4a) are separated into mean contribution and local perturbation contribution by surface horizontal wind speed and thermodynamic variables. Assuming $X$ is a temporally and spatially varying variable, we define $X = \{X\} + X'$, where $\{X\}$ indicates the horizontal mean (latitude and longitude) and $X'$ the anomaly from the horizontal mean, and $X = [X] + X^*$, where $[X]$ indicates the time mean and $X^*$ the anomaly from the time mean. If a field $Y$ is a product of $X_1$ and $X_2$, i.e., $Y = X_1 X_2$, then $\{Y\} = \{X_1\}\{X_2\} + \{X_1' X_2'\}$ for the longitudinal mean (similarly for the time mean).

In the turbulence scheme used in ICON (Raschendorfer, 2001; Mellor and Yamada, 1982), the turbulent exchange coefficients are proportional to the turbulent kinetic energy, and so we expect the coefficients to depend on surface wind speed (as well as vertical stability). This creates an overall more than linear dependence of the surface fluxes on wind speed. For simplicity, we combine the coefficients and surface wind speed together, i.e., $c_E \overline{U}_h$ for the surface latent heat flux and $c_H \overline{U}_h$ for the surface sensible heat flux. Here, we derive turbulent exchange coefficients from the other variables in Eq. (3). For simplicity, we set the air density in Eq. (3) to a constant value of $1.2\,\text{kg}\,\text{m}^{-2}$. Then, the surface latent and sensible heat fluxes vary with $c_E \overline{U}_h$ and $\Delta q$, and $c_H \overline{U}_h$ and $\Delta T$, respectively. For example, a zonal mean of the surface latent heat flux can be expressed by the longitudinal mean and its fluctuation as $\{F_{\text{latent heat}}\} = \rho_s L_v \{c_E \overline{U}_h\}\{\Delta q\} + \rho_s L_v \{(c_E \overline{U}_h)' \Delta q'\}$. Thus, the zonal and

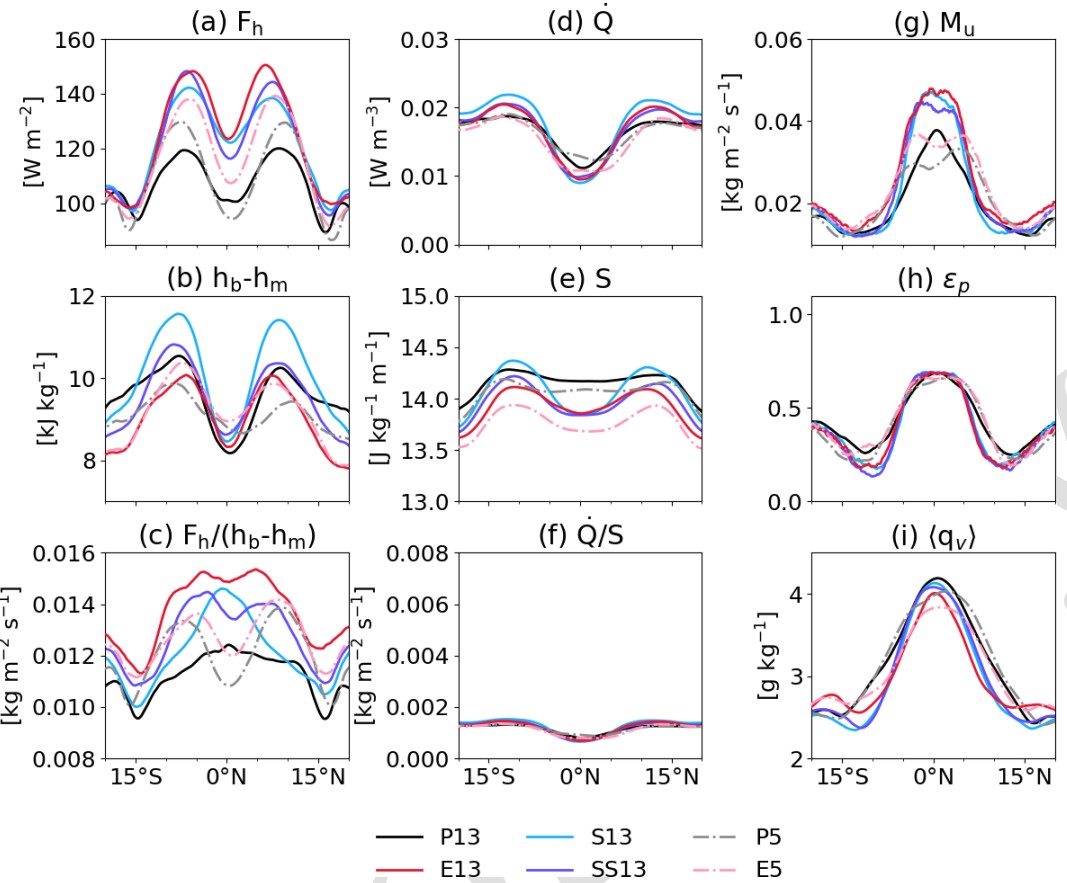

**Figure 4.** Time and zonal mean of (**a**) the surface enthalpy flux, (**b**) the vertical difference in moist static energy, (**c**) the ratio of the surface enthalpy flux and the vertical difference in moist static energy, (**d**) the lower tropospheric radiative cooling, (**e**) the dry static stability, (**f**) the ratio of the lower tropospheric radiative cooling and dry static stability, (**g**) estimated convective mass flux, (**h**) estimated precipitation efficiency, and (**i**) the column averaged specific humidity. Ranges of the *y* axes in panels (**c**) and (**f**) are identical to facilitate comparison.

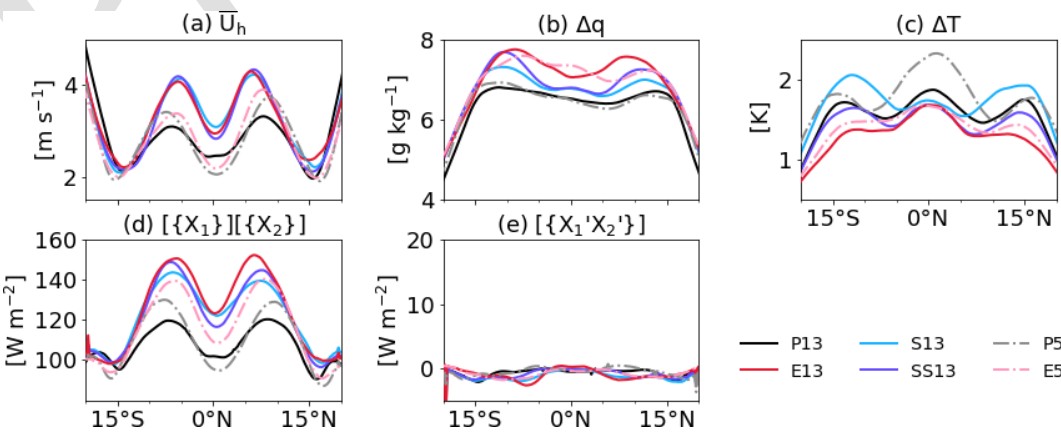

**Figure 5.** Time and zonal mean of surface properties: (**a**) the surface horizontal wind speed, (**b**) specific humidity contrast between the air and the ocean surface, and (**c**) temperature contrast between the air and the ocean surface. The bottom panels show contributions of (**d**) mean terms and (**e**) covariance terms (Eq. 4) to time and zonally averaged $F_h$. Here $X$ represents the components of $F_h$ such as $c_E \overline{U}_h$, $c_H \overline{U}_h$, $\Delta q$ and $\Delta T$.

time mean of $F_\text{h}$ can be expressed as

$$[\{F_\text{h}\}] = \rho_s L_\text{v}\big[\{c_\text{E}\overline{U}_\text{h}\}\big]\big[\{\Delta q\}\big] + \rho_s c_\text{p}\big[\{c_\text{H}\overline{U}_\text{h}\}\big]\big[\{\Delta T\}\big]$$
$$+ \rho_s L_\text{v}\big[\{(c_\text{E}\overline{U}_\text{h})'\Delta q'\}\big] + \rho_s c_\text{p}\big[\{(c_\text{H}\overline{U}_\text{h})'\Delta T'\}\big]$$
$$+ \rho_s L_\text{v}\big[\{c_\text{E}\overline{U}_\text{h}\}^*\{\Delta q\}^*\big] + \rho_s c_\text{p}\big[\{c_\text{H}\overline{U}_\text{h}\}^*\{\Delta T\}^*\big]. \quad (4)$$

The first and second terms on the right-hand side in each row are from the surface latent and sensible heat fluxes, respectively. In the first row $[\{X_1\}][\{X_2\}]$ indicates the contributions of time and zonally averaged surface wind speed combined with the turbulent coefficients and thermodynamic effects to the time and zonal mean of $F_\text{h}$. In the second row $[\{X_1'X_2'\}]$ indicates a product of local fluctuations which is averaged over time and longitude, i.e., the covariance which indicates the contributions of local perturbations. In the last row $[\{X_1\}^*\{X_2\}^*]$ indicates the time mean of a product of temporal fluctuations of spatial mean. Equation (4) represents fields that are averaged over longitude first and then time. Averaging, which is carried out over time and then longitude, is also tested (not shown) and does not change the results that are shown below.

Figure 5d and e show latitudinal variations of contributions of the time-zonal mean values and the covariance terms (the terms in the first and second rows of Eq. 4, respectively). Overall, the contributions of the time-zonal mean values (Fig. 5d) match the patterns of the actual $F_\text{h}$, while the covariance terms (Fig. 5e) fluctuate around zero. The temporal anomalies of the zonal mean components (the terms in the last row of Eq. 4) are very small (not shown), compared to the mean and covariance terms. Therefore, the decomposition analysis indicates that the time-zonal mean of $c_\text{E}\overline{U}_\text{h}$ and $\Delta q$ is the main contributor to shape the mean $F_\text{h}$ for all experiments.

In summary, the mean surface properties (Fig. 5a and b) are dominant over the covariance and anomaly terms to shape the mean $F_\text{h}$ (Fig. 4a). At 13 km, the substantial differences between explicit and parameterized deep convection are found in the surface wind speed (Fig. 5a), which shape the main differences in surface enthalpy fluxes (Fig. 4a). The variations in the moisture contrast (Fig. 5b) create additional minor differences in surface enthalpy fluxes, e.g., among E13, S13 and SS13. For these coarse-resolution runs, surface enthalpy fluxes are controlled by surface winds, which are in fact closely coupled to the large-scale circulation (Fig. 3a–d), which becomes stronger with explicit deep convection. The resolution dependence exhibits similar relations among surface enthalpy fluxes, surface winds and large-scale circulation between P13 and P5 and between E13 and E5. The changes in the moisture contrast are subtle due to the horizontal resolution, except for E5 that reduces moisture contrast in the Northern Hemisphere, which offsets the asymmetry in the local maxima of surface enthalpy fluxes there. Surprisingly, the links we discussed so far do not apply to the sensitivity to the treatment of convection at 5 km. The surface wind speed (Fig. 5a) is similar between E5 and P5. Meanwhile, the moisture contrast (Fig. 5b) increases from P5 to E5, mainly in the inner tropics, contributing to the enhanced surface fluxes in E5. This indicates that a modest change in resolution substantially alters the relation between surface fluxes and surface properties due to convective treatment.

### 5.1.2 Statistical distribution

Previously, the latitudinal distributions of mean $F_\text{h}$ were examined. We here construct statistical distributions of surface horizontal wind speed, thermodynamic disequilibrium and surface fluxes (Hsu et al., 2022) to provide a complementary view. This does not require considering the turbulent exchange coefficients and allows us to examine how dependent surface enthalpy fluxes are on surface wind speed and thermodynamic disequilibrium. Specifically, we ignore the surface sensible heat flux, which accounts for only about 10 % of surface enthalpy fluxes, and focus on the surface latent heat flux. Surface latent heat flux is grouped by bins of $\overline{U}_\text{h}$ and $\Delta q$ to outline distributions of the variables and the surface flux in one figure. We sample the surface latent heat flux by bins of $\overline{U}_\text{h}$ and $\Delta q$ at every output time step of 1 h and in every 0.2° latitude–longitude grid box. The bin size for sampling is $1\,\text{m\,s}^{-1}$ for $\overline{U}_\text{h}$ and $1\,\text{g\,kg}^{-1}$ for $\Delta q$ as in Hsu et al. (2022). We focus on the area between 10° N and 10° S where large differences in surface enthalpy fluxes are observed (Fig. 4a). The results, however, do not change much when considering the area between 20° N and 20° S.

Figure 6 depicts two-dimensional histograms of $\overline{U}_\text{h}$ and $\Delta q$, as well as corresponding values of the surface latent heat flux (contour). For P13, the density distributions of both $\overline{U}_\text{h}$ and $\Delta q$ are positively skewed with an extensively long tail for the former (Fig. 6a). The bin of $\overline{U}_\text{h}$ of $1$–$2\,\text{m\,s}^{-1}$ and $\Delta q$ of $6$–$7\,\text{g\,kg}^{-1}$ contains the maximum frequency density of 15 % (colored dot). Contour lines demonstrate that the corresponding surface latent heat flux is more strongly dependent on $\overline{U}_\text{h}$ than on $\Delta q$. The maximum frequency density (colored dot) is located between the contour lines of $50$–$100\,\text{W\,m}^{-2}$. A similar pattern is observed for P5 (Fig. 6c). However, the maximum frequency density is slightly reduced to 12.5 %, the $\Delta q$ distribution is broader with the upper limit extending to $12\,\text{g\,kg}^{-1}$ and the tail of the $\overline{U}_\text{h}$ distribution becomes shorter, compared to P13. This shorter tail may be associated with downward momentum transport in the ITCZ (Fig. 3a and e).

E13 (Fig. 6b) exhibits the largest contrast to P13, showing relatively evenly distributed $\overline{U}_\text{h}$ and $\Delta q$. The maximum density (colored dot) accounts for 6.0 %, which is less than half of that for P13 in the bin of $\overline{U}_\text{h}$ of $3$–$4\,\text{m\,s}^{-1}$ and $\Delta q$ of $7$–$8\,\text{g\,kg}^{-1}$, showing a greater surface wind speed and greater moisture contrast (seen also in Fig. 5a and b). The surface latent heat flux (contour lines) increases strongly with increasing $\overline{U}_\text{h}$, while to a lesser extent but noticeably it increases with growing $\Delta q$. As expected from the high $F_\text{h}$ shown in Fig. 4a, the maximum density bin is located between the con-

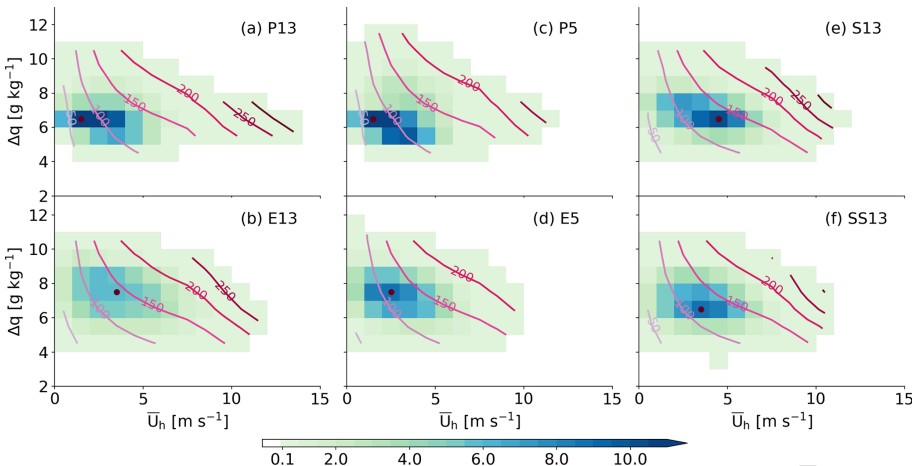

**Figure 6.** Two-dimensional histogram of surface wind speed and moisture contrast between the ocean surface and atmosphere in density (shading) with dots indicating the maxima. The minimum frequency density is 0.1 for shading. Contour lines indicate corresponding surface latent flux [$W\,m^{-2}$] binned by the wind speed and moisture contrast. The contour interval is $50\,W\,m^{-2}$, increasing from lighter to darker colors.

tour lines of $100–150\,W\,m^{-2}$. The histogram for E5 (Fig. 6d) shares similarities to that for E13. However, the $\overline{U}_h$ distribution is restricted to $10\,m\,s^{-1}$, and the upper limit for $\Delta q$ is extended to $12\,g\,kg^{-1}$ as seen in the differences between P13 and P5. The maximum frequency density increases to 8.8 % in the bin of $\overline{U}_h$ of $2–3\,m\,s^{-1}$ and $\Delta q$ of $7–8\,g\,kg^{-1}$.

The distributions of S13 and SS13 (Fig. 6e and f) show some intermediate features between E13 and P13. The maximum frequency density (colored dot) lies between the contour lines of $100–150\,W\,m^{-2}$, consistent with small differences in $F_h$ among the explicit deep convection runs (Fig. 4a). The distributions are concentrated at the highest frequency density of 10 % for S13 and 8.4 % for SS13, which are both in the $\Delta q$ bin of $6–7\,g\,kg^{-1}$ as in P13.

## 5.2 Vertical difference in moist static energy

Figure 4b shows the vertical difference in $h$ between the BL and the lower troposphere described by $h_b - h_m$ (Eq. 1). This contrast is key for BLQE, as it determines the reduction of $h$ in the BL through convective downdrafts and large-scale subsidence. In the moist ITCZ region $h_b - h_m$ has minimum with relatively small differences among the six simulations and P13 showing the smallest values. Then $h_b - h_m$ increases markedly in the trade wind belt with a much larger dependence on convective treatment and resolution, followed by a gradual falloff towards higher latitudes. In the trade wind area, $h_b - h_m$ is smallest for E13 among the coarse-resolution runs, indicating deep mixing and conditions closer to moist neutrality. S13 shows much increased contrasts, suggesting that here deep mixing may be suppressed at the cost of more subtle shallow mixing. SS13 lies in the middle between these two extremes. P13 shows a fundamentally different behavior with a much slower falloff towards higher latitudes. The

high-resolution runs show similar patterns of $h_b - h_m$ as in the coarse-resolution runs, but with relatively smoother distributions in $h_b - h_m$ in the inner tropics. In the ITCZ, $h_b - h_m$ slightly increases from 13 to 5 km grid spacing. In the outer tropics, P5 exhibits overall smaller $h_b - h_m$ than P13, while $h_b - h_m$ for E5 closely matches that for E13 but with a more pronounced asymmetry between the hemispheres.

Profiles of $h$ provide a deeper insight into the $h_b - h_m$ patterns. The solid and dash-dotted lines in Fig. 7 show $h$ profiles below 8 km at characteristic latitudes for the ITCZ (0°), trades (8° N/S) and subsidence areas (15° N/S). Overall, $h$ shifts to lower values from the Equator to higher latitudes, following the prescribed SST pattern. The dotted lines in Fig. 7 show corresponding profiles of dry static energy ($s = \phi + c_p T$) with hardly any difference between the experiments. Therefore, lower $h$ with increasing latitude is largely equivalent to drier air.

First, we discuss $h$ profiles for the coarse-resolution runs in the ITCZ (Fig. 7a). The BL top for explicit deep convection (E13, S13 and SS13) is at 500 m but shallower for P13 ($\sim 400$ m or one model level lower), while $h_b$ differs little, changing by 0.1 %–0.3 % only (see Table 2). Note that the BL height is fixed for our diagnostic tool to the layer of 10–500 m, but calculating $h_b$ with alternating BL heights below or slightly above 500 m does not impact the results. Among the coarse-resolution runs, E13 shows the lowest value of $h_b$, possibly related to more frequent and more intense convective downdrafts in line with more intense rainfall (Fig. 2b). In the lower free troposphere, more distinct differences are evident, specifically between 1 and 3 km. E13 has again the overall lowest values, such that downdrafts can more effectively reduce $h_b$. Retsch et al. (2019) also found a drier lower troposphere for their explicit deep convection cases than for parameterized ones. S13 and SS13 show enhanced values rel-

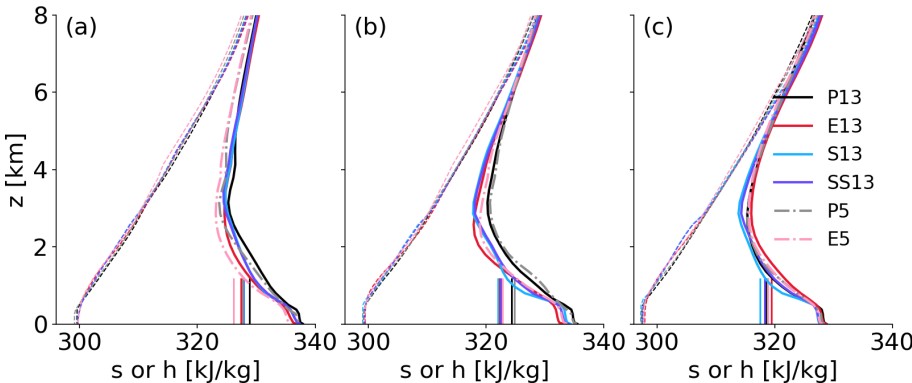

**Figure 7.** Profiles of the time and zonal mean of dry static energy (dashed) and moist static energy (solid and dash-dotted) [kJ kg$^{-1}$] at **(a)** 0°, **(b)** 8° N/S and **(c)** 15° N/S. Those latitudes are chosen since they are representative of the ITCZ, the trade wind belt and the subsiding areas (see Sect. 3). The vertical bars represent the calculated $h_m$ (average over 0.5–5 km).

**Table 2.** Time and zonal average of moist static energy [kJ kg$^{-1}$] and dry stability [J kg$^{-1}$ m$^{-1}$] at three different latitudes. The layer of 10–500 m and 0.5–5 km are used for the quantities in the BL $h_b$ and in the lower troposphere $h_m$ and dry stability $S$, respectively.

| Lat (°) | Exp. name | $h_b$ | $h_m$ | $h_b - h_m$ | $S$ |
|---|---|---|---|---|---|
| 0 | P13 | 337.08 | 328.88 | 8.20 | 14.17 |
| | E13 | 335.70 | 327.37 | 8.33 | 13.86 |
| | S13 | 336.34 | 327.88 | 8.45 | 13.85 |
| | SS13 | 336.37 | 327.75 | 8.62 | 13.84 |
| | P5 | 336.51 | 327.70 | 8.81 | 14.09 |
| | E5 | 335.09 | 326.14 | 8.96 | 13.69 |
| 8 | P13 | 334.65 | 324.28 | 10.37 | 14.22 |
| | E13 | 332.16 | 322.16 | 10.00 | 14.07 |
| | S13 | 333.52 | 322.05 | 11.48 | 14.25 |
| | SS13 | 333.06 | 322.48 | 10.58 | 14.08 |
| | P5 | 334.36 | 324.81 | 9.55 | 14.09 |
| | E5 | 332.90 | 322.80 | 10.10 | 13.85 |
| 15 | P13 | 328.10 | 318.46 | 9.64 | 14.20 |
| | E13 | 327.96 | 319.42 | 8.54 | 13.92 |
| | S13 | 327.27 | 317.51 | 9.76 | 14.17 |
| | SS13 | 327.34 | 318.22 | 9.12 | 14.06 |
| | P5 | 327.73 | 318.74 | 8.99 | 14.14 |
| | E5 | 327.42 | 318.81 | 8.61 | 13.82 |

ative to E13 around 2 km, while P13 has higher $h$ throughout most of the layer up to 5 km. Applying a vertical average over 0.5 to 5 km, we obtain $h_m$, which varies between 327.4 and 328.9 kJ kg$^{-1}$ (Table 2). Given that differences within and above the BL are largely consistent between the coarse-resolution runs, $h_b - h_m$ in the ITCZ increases by 3 %–6 % from P13 to S13, SS13 and E13, as also evident from Fig. 4b.

At 5 km, changes in $h$ profiles due to convective treatment are largely consistent with the results at 13 km (Fig. 7a), showing that E5 produces lower $h$, specifically above the BL, and a higher BL top in E5 (500 m) than in P5 (270 m or two

model levels lower). The drier $h$ in the lower troposphere is the main reason for the increased $h_b - h_m$ in E5 compared to P5 (Fig. 4b). When we switch our focus to resolution dependence (P13 versus P5 or E13 versus E5), the higher resolution dries the atmospheric column, which is consistent with a drier ITCZ (Fig. 2a). This lower $h$ is more pronounced in the lower troposphere than in the BL, leading to reduced $h_m$ for P5 and E5 than for P13 and E13 (Table 2). Thus, $h_b - h_m$ in the ITCZ is larger for the high-resolution runs (Fig. 4b).

In the trade wind belt, differences in $h_b - h_m$ among the experiments are larger than in the ITCZ (Fig. 4b). This corresponds with more marked differences in the vertical profiles of $h$ (Fig. 7b). At 13 km, $h$ profiles are shifted to lower values for the explicit deep convection runs (E13, S13 and SS13) than P13. The variations of $h_b$ among the coarse-resolution experiments are systematic, decreasing from P13 to S13 or SS13 to E13 by 1.13–2.49 kJ kg$^{-1}$ (see Table 2). Meanwhile, $h_m$ varies little among the explicit deep convection runs (E13, S13 and SS13; colored vertical bars in Fig. 7b) but decreases from P13 to other coarse-resolution runs by 1.8–2.23 kJ kg$^{-1}$ (Table. 2). SS13 exhibits an intermediate behavior of $h$ between S13 and E13, such that SS13 is similar to S13 in the BL, to E13 between 0.5–1.5 km and again to S13 above 1.5 km. This intermediate behavior indicates that the stochastic scheme for shallow convection (SS13) mixes the air between the BL and lower troposphere more efficiently than the deterministic version (S13) but not as deeply as the explicit one (E13). Consequently, this results in some unsystematic behavior of $h_b - h_m$ from one to another, with E13 showing the lowest value, then P13 and SS13, and finally with S13 showing the highest value (Fig. 4b).

For the high-resolution runs, the effect of convective treatment on the $h$ profile is consistent with that for the coarse-resolution runs, again with the drier $h$ profile for E5 than for P5 in the trade wind belt (Fig. 7b). Compared to the coarse-resolution runs, both P5 and E5 moisten the lower troposphere, which may be associated with the broad ITCZ

(Fig. 2a), while the BL is moistened only for E5 possibly through convective mixing, which transports higher $h$ into the BL. Despite the coherent effect of convective treatment on the $h$ profile between the coarse- and high-resolution runs, the resulting $h_b - h_m$ (Fig. 4b) decreases from parameterized to explicit deep and shallow convection at 13 km but increases at 5 km.

In the subsidence region (Fig. 7c), the BL height is 500 m in all experiments, and $h_b$ is fairly similar across the runs (Table. 2). Among the coarse-resolution runs, E13 shows the largest value of $h_m$, indicating more moisture columns than the others as profiles for dry static stability show no substantial differences. This is the opposite of the results by Retsch et al. (2019), who showed that the lower troposphere in the subsidence region is drier with explicit deep convection than parameterized deep convection. Given that the strength of the Hadley circulation is largely comparable between the runs with explicit deep convection (discussed in Sect. 3.2), lower-tropospheric moisture is presumably dominated by local mixing rather than large-scale subsidence effects. Likewise, S13 and SS13 fundamentally differ in that vertical mixing, mainly in the lower atmospheric layer, is more efficient for the stochastic version (SS13) (Sakradzija et al., 2020). Accordingly, $h_m$ for SS13 increases from S13 by more than 0.5 kJ kg$^{-1}$, which is less than 0.5 kJ kg$^{-1}$ in other latitudinal regions (Table. 2). Thus, we see a systematic change of $h_b - h_m$ in the subsidence region, representing some effects of local mixing. The high-resolution runs show visually identical $h$ profiles (Fig. 7c) with small differences in $h_b$ and $h_m$ (Table 2).

In summary, $h_b$ and $h_m$ in the ITCZ and the trade wind belt show some systematic changes due to convective treatment, with $h$ profiles in the former being more sensitive to horizontal resolution. However, subtle changes in $h_b$ and $h_m$ result in complex patterns of $h_b - h_m$. The impact of shallow convective treatment is evident in profiles of $h$, particularly in the trade wind belt. The stochastic version of shallow convection (SS13) exhibits an intermediate behavior of $h$ between the deterministic (S13) and explicit version (E13), reflected in the latitudinal distribution of $h_b - h_m$ (Fig. 4b).

## 5.3 Radiative cooling

Figure 4d shows the time and zonal mean of radiative cooling in the lower troposphere (0.5–5 km). In P13 and P5 the latitudinal distribution of $\dot{Q}$ is overall the flattest, with it marking its minimum in the ITCZ and larger values in the outer tropics (Fig. 4). E13, S13, SS13 and E5 show a similar pattern but less cooling in the ITCZ. The explicit deep convection runs have stronger cooling in the outer tropics, which is particularly true for S13. One exception is E5 which is more consistent with the parameterized deep convection runs beyond 10° N/S. In the following, we discuss this result in the context of total cloud cover (Fig. 8) and the latitudinal-height distribution of the radiative temperature tendency (Fig. 9).

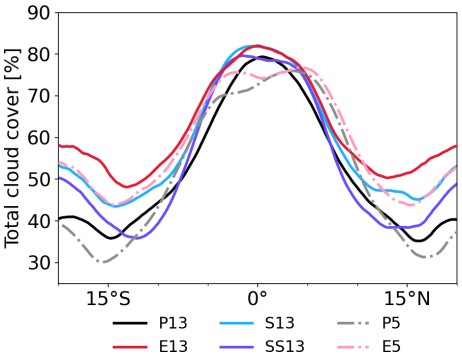

**Figure 8.** As in Fig. 4d but for total cloud cover.

Figure 8 shows that amongst all runs, E13 has the overall highest cloud cover, peaking at about 80 % at the Equator and falling off gradually to about 50 % at around 15° N/S, beyond which there is a slight increase again. Consequently, E13 exhibits net radiative cooling in the troposphere with a marked contrast between the ITCZ region and the outer tropics (Fig. 9b). In the ITCZ, radiative cooling is generally reduced, and in the outer tropics there are signatures of shallow and congestus clouds, forming a trimodal structure (Johnson et al., 1999; Khairoutdinov et al., 2009), consistent with the highest cloud cover relative to the other runs (Fig. 8).

P13 show the largest contrast to E13 with reduced cloud cover, especially in the outer tropics (Fig. 8), and with some differences in the pattern of radiative cooling (Fig. 9a). In the ITCZ, radiative cooling is generally reduced, and there is even a slight warming below the tropopause, likely related to longwave absorption by optically thick cirrus (Senf et al., 2020). This is consistent with the fact that for P13 cloud ice in the upper troposphere is spread over a 1 km deeper layer than in the other coarse-resolution experiments (not shown). Note that the near-tropopause warming is not included in $\dot{Q}$ (Fig. 4d), which is averaged over 0.5–5 km. In the outer tropics, radiative cooling increases and is quite homogeneous across most of the free troposphere, decreasing gently above about 9 km with no indication of the trimodal structure (Fig. 9a). The top of the BL stands out as an area of enhanced cooling associated with longwave emission from the top of shallow clouds into the relatively dry free troposphere above it.

S13 exhibits similar features to E13 in terms of cloud cover and radiative cooling (including $\dot{Q}$) in the ITCZ, but there are marked differences in the outer tropics. Cloud cover in S13 (47.7 %) is intermediate between P13 (39.8 %) and E13 (53.2 %) (Fig. 8). While the free-tropospheric cooling by radiation is consistent with E13, S13 (Fig. 9c) reveals that radiative cooling above the BL is substantially enhanced and very concentrated, creating a gap in cooling above that. This leads to the largest $\dot{Q}$ of all runs in the outer tropics (Fig. 4d). SS13 shows the cloud cover closest to P13 among the other coarse-resolution runs (Fig. 8). Free-tropospheric radiative

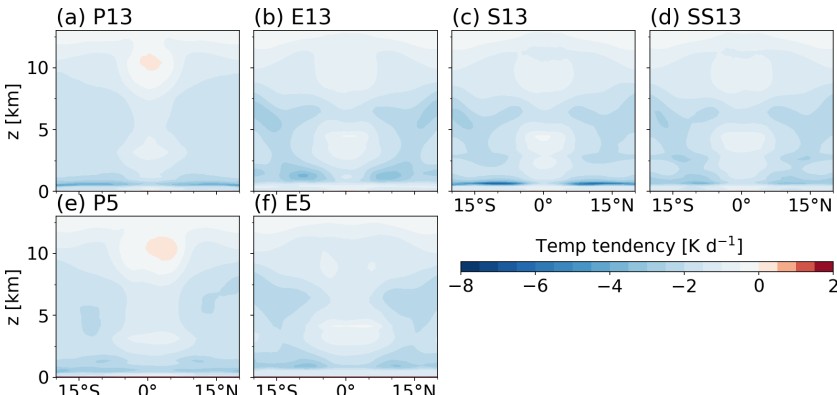

**Figure 9.** As in Fig. 3 but for net radiative temperature tendency.

cooling for SS13 (Fig. 9d) remains similar to S13 and E13, with slightly enhanced cooling at around 3 km, but the BL top cooling in the outer tropics is much reduced. This is due to the fact that the stochastic version (SS13) allows for effi-
5 cient mixing between the BL and the lower troposphere and for efficient BL convective heating (Sakradzija et al., 2020; Senf et al., 2020). Consequently, this leads to overall similar $\dot{Q}$ to E13 (Fig. 4d), despite the different vertical structures of radiative temperature tendency.
The high-resolution runs (P5 and E5) show differences in total cloud cover and radiative cooling, largely consistent with the differences between P13 and E13. The total cloud cover is reduced from E5 to P5, particularly in the outer tropics (Fig. 8). P5 is characterized by net warming below the
tropopause (Fig. 9e), which is slightly shifted to the Northern Hemisphere where the maximum mean rainfall is (Fig. 2a). The distribution of radiative cooling for E5 (Fig. 9f) exhibits a trimodal structure, but net cooling is generally weaker than that for E13, leading to reduced $\dot{Q}$ for E5 in the outer tropics
compared to other explicit deep convection runs (Fig. 4d).

### 5.4 Dry stability

In the conceptual framework, the effects of radiative cooling need to be considered relative to the dry stability $S$ (Eq. 1), which is shown in Fig. 4e. For P13 and P5, $S$ is nearly
25 constant between 15° N TS1 and 15° S with a value of ~~14.0 and 13.9~~ J kg$^{-1}$ m$^{-1}$ TS2, respectively, beyond which it drops slightly. The other experiments exhibit some noteworthy differences. In the ITCZ, explicit deep convection appears to create a somewhat lower stability by ~~0.3–0.4~~ J kg$^{-1}$ m$^{-1}$ TS3
compared to parameterized deep convection at the same resolution (Table 2). This may be related to the fact that the convection scheme triggers convection a little more easily and can therefore more effectively stabilize the atmosphere in a convectively active region. This subtle difference is also ev-
ident from Fig. 7a. (The slope of $s$ in the lower troposphere represents $S$.) In the outer tropics, the treatment of shallow convection has some effect on $S$, increasing $S$ by about

0.1 J kg$^{-1}$ m$^{-1}$ from one experiment to another (Table 2): E13 likely produces the least shallow mixing and is thus least stable, followed by SS13 and S13. Finally, the largest
stability is found in P13, which parameterizes both shallow and deep convection. With increasing resolution from 13 to 5 km, $S$ decreases systematically. As in the ITCZ, differences are overall subtle, which is also evident from profiles of $s$ (Fig. 7b and c).
In the conceptual model, the ratio of radiative cooling in the lower troposphere $\dot{Q}$ and dry stability $S$ are considered (second term in Eq. 1). As radiation and stability in all runs largely compensate for each other, the ratio is almost identical and also relatively constant over latitudes (Fig. 4f). In
contrast, the effect of surface enthalpy fluxes and vertical contrast of $h$ leads to substantial variations in the first term in Eq. (1) (Fig. 4c). This demonstrates that despite all the details discussed above, the thermodynamic balance between radiation and stability hardly drives differences between the
simulations.

### 5.5 Convective updraft mass flux and precipitation efficiency

Up to this point, we have examined the distributions of each term on the right-hand side in Eq. (1). To conclude our di-
60 agnostic framework, here we discuss $M_u$, $\epsilon_p$ and $\langle q_v \rangle$, which are directly related to rainfall (Eq. 2), and then link this discussion to the entire framework. We start with quantities in the ITCZ. Surprisingly, $\epsilon_p$ (Fig. 4h TS4) is quite similar across the runs with a maximum value of 0.63–0.657 (Ta-
65 ble 3). Note that the time-averaged quantities are taken into account here, but timely varying Pr and $\epsilon_p$ can be strongly correlated (Narsey et al., 2019; Muller and Takayabu, 2020). Furthermore, $\epsilon_p$ can depend significantly on how convection is treated in models (Li et al., 2022), but the different con-
70 vective treatments do not alter $\epsilon_p$ in the ITCZ in our case. Meanwhile, $\langle q_v \rangle$ (Fig. 4i) marginally decreases from parameterized to explicit convection and from the coarse to high resolutions (Table 3), indicating that P13 has the moistest

atmosphere and E5 is the driest in the ITCZ, as also seen in profiles of $h$ (Fig. 7a). Given the almost identical $\epsilon_p$ and the marginal changes in $\langle q_v \rangle$, $M_u$ must substantially differ among the runs to match the differences in rainfall (Eq. 2). Correspondingly, the latitudinal distribution of $M_u$ (Fig. 4g) closely matches that of mean rainfall (Fig. 2a). For example, the coarse-resolution runs show that $M_u$ in the ITCZ increases from parameterized to explicit deep convection by 30 %–38 % (Table. 3), which is largely consistent with the mean rainfall increase (about 35 %). For the high-resolution runs, $M_u$ values increase by 16 % from P5 to E5, which is larger than the mean rainfall increase of 11 %. This gap is noticeable at around 3° N where the mean rainfall increases by 3 % from P5 to E5 (Fig. 2a), while $M_u$ increases by 10 % (Fig. 4g). This is in fact compensated for by the decrease of 6 % in $\langle q_v \rangle$ (Fig. 4i).

In the outer tropics, $\epsilon_p$ sharply decreases, reaches a minimum at around 9–12° N/S and beyond this increases again with latitude (Fig. 4h). The differences among the runs are substantial in the outer tropics, with $\epsilon_p$ varying between 0.221 and 0.310 (highest in P13). Mean values of $\langle q_v \rangle$ slightly vary, with P13 marking the greatest value of 2.77 g kg$^{-1}$ (Fig. 4i). From the ITCZ to the outer tropics $M_u$ sharply decreases to about 0.015 kg m$^{-2}$ s$^{-1}$ and beyond the minimum slightly increases again with latitude (Fig. 4g). These patterns with a minimum and marginal increase in $\epsilon_p$ and $M_u$ are also observed in Pr (Fig. 2a), yet the differences in Pr in the outer tropics do not vary as much as those in $\epsilon_p$ or $\langle q_v \rangle$ due to the generally low values of the contributing variables there.

We summarize the results by focusing on the ITCZ where mean rainfall varies most due to the treatment of convection and horizontal resolution. Our results indicate that in the ITCZ, the difference in $M_u$ from one experiment to another most closely matches that in Pr, with almost identical $\epsilon_p$ and marginal changes in $\langle q_v \rangle$. When revisiting all input variables in Eq. (1) for our diagnostic tool (Fig. 4), $M_u$ is shaped by $F_h/(h_b - h_m)$, which describes BLQE, while variations in $\dot{Q}$ and $S$ largely compensate for each other (Fig. 4c and f). Given almost constant $\epsilon_p$ in the ITCZ, an increase in $M_u$ increases $M_d$, through $M_d = (1 - \epsilon_p)M_u$, which carries low $h$ from the lower troposphere into the BL to balance enhanced $F_h$. At 13 km, the treatment of deep convection produces the main differences in $M_u$ by 30 %–38 % between the explicit and parameterized versions, which is associated with substantial differences in $F_h$ (20 %–28 %). This change in $F_h$ is associated with changed $\overline{U}_h$ (Sect. 5.1), which in turn is closely linked to the Hadley circulation (Sect. 3.2). Note that these links are not unidirectional but multidirectional interactions in the sense that we cannot disentangle whether a stronger circulation leads to more rainfall or vice versa. From 13 to 5 km, the results share similar importance for mean rainfall, which is again controlled by $M_u$. Decreased $M_u$ with increasing resolution is associated with the combined effects of increased $h_b - h_m$ and suppressed $F_h$, which is shaped by

$\overline{U}_h$ through the large-scale circulation. This indicates that the thermodynamics at low altitudes become important at higher resolution. The difference in $M_u$ between P5 and E5 is again associated with that in $F_h$, which in fact changes due to $\Delta q$ rather than $\overline{U}_h$. Furthermore, a marginal change in $\langle q_v \rangle$ has a small contribution to mean rainfall, showing again the importance of thermodynamic properties at low altitudes because substantial changes in $\langle q_v \rangle$ are found below 6 km, inferring from $h$ profiles (Fig. 7).

## 5.6 Meridional advection

Equation (1) is obtained by neglecting the horizontal advection in the BL. Here we test the sensitivity of the diagnostic tool when the advection term is included. A scale analysis for $h$ budget of the BL (Eq. A3) reveals that the BL radiative cooling term can safely be ignored while the advection term is not fully negligible ($\mathrm{d}\dot{Q}_b \sim 1\,\mathrm{W\,m^{-2}}$, $\mathrm{d}\rho\mathbf{V}_h \cdot \nabla h \sim 10\,\mathrm{W\,m^{-2}}$, $F_h \sim 100\,\mathrm{W\,m^{-2}}$). With this, Eq. (1) can be expressed as

$$M_u = \frac{1}{1 - \epsilon_p}\left( \frac{F_h - \mathrm{d}\rho\mathbf{V}_h \cdot \nabla h}{h_b - h_m} - \frac{\dot{Q}}{S} \right), \tag{5}$$

where d is the BL height and $\mathbf{V}_h$ is the horizontal velocity. Here, only meridional advection is taken into account because BL meridional and zonal gradients are in the order of 10 and 0.1 J kg$^{-1}$ km$^{-1}$, respectively, while BL meridional and zonal winds are comparable in magnitude. The advection term is calculated by integrating the meridional advection of $h$ from the lowest atmospheric layer at 10 m to the BL top at 500 m and assuming an air density of 1.2 kg m$^{-2}$.

Figure 10 shows the latitudinal distribution of BL meridional advection and the impact of including it on precipitation efficiency and convective updraft mass flux. Parameterized deep and shallow convection (Fig. 10a) produces subtle meridional advection in the BL with near-zero values at the Equator that increases away from the Equator, leading to an average of 4.4 and 3.3 W m$^{-2}$ in the ITCZ for P13 and P5, respectively (Table 3). The small advection term in the ITCZ decreases $M_u$ by about 0.0005 kg m$^{-2}$ s$^{-1}$ and increases $\epsilon_p$ by about 0.01 compared to the advection-free diagnostic tool (Table 3). At around 10° N/S, BL meridional advection reaches a maximum of 14 W m$^{-2}$ and beyond that decreases with increasing latitude, leading to a small increase in $\epsilon_p$ by 0.02 and a small decrease in $M_u$ by 0.001 kg m$^{-2}$ s$^{-1}$.

In contrast to parameterized deep and shallow convection (P13 and P5), E13 exhibits the largest contribution of BL meridional advection (Fig. 10a). In the ITCZ, the averaged advection is 27.3, consequently reducing $M_u$ by 0.003 kg m$^{-2}$ s$^{-1}$ but increasing $\epsilon_p$ by 0.05 (Table 3). At around 7° N/S the advection term shows local maxima of 32.8 W m$^{-2}$ and then sharply decreases with increasing latitude. The overall large meridional advection term for E13 is consistent with intensified $\overline{U}_h$ (Fig. 5a) and a greater $h$ change in the BL with latitude (Fig. 7). While the advection

**Table 3.** The averaged precipitation rate (Pr), column specific humidity ($\langle q_v \rangle$), convective updraft mass flux ($M_u$), precipitation efficiency ($\epsilon_p$), surface enthalpy flux ($F_h$) and BL meridional advection (Adv) between 5° N and 5° S for each experiment. The quantities in parentheses indicate those when the BL meridional advection is included (Eq. 5).

| | Pr [mm d$^{-1}$] | $\langle q_v \rangle$ [g kg$^{-1}$] | $M_u$ [kg m$^{-2}$ s$^{-1}$] | $\epsilon_p$ | $F_h$ [W m$^{-2}$] | Adv [W m$^{-2}$] |
|---|---|---|---|---|---|---|
| P13 | 7.2 | 4.0 | 0.0318 (0.0313) | 0.633 (0.643) | 105.6 | 4.4 |
| E13 | 9.76 | 3.74 | 0.0439 (0.0409) | 0.648 (0.698) | 134.9 | 27.3 |
| S13 | 9.76 | 3.91 | 0.0416 (0.0393) | 0.657 (0.696) | 129.3 | 20.9 |
| SS13 | 9.64 | 3.92 | 0.0412 (0.0389) | 0.657 (0.696) | 127.0 | 20.7 |
| P5 | 6.72 | 3.92 | 0.0302 (0.0298) | 0.630 (0.638) | 103.2 | 3.3 |
| E5 | 7.5 | 3.72 | 0.035 (0.0335) | 0.643 (0.672) | 118.3 | 13.9 |

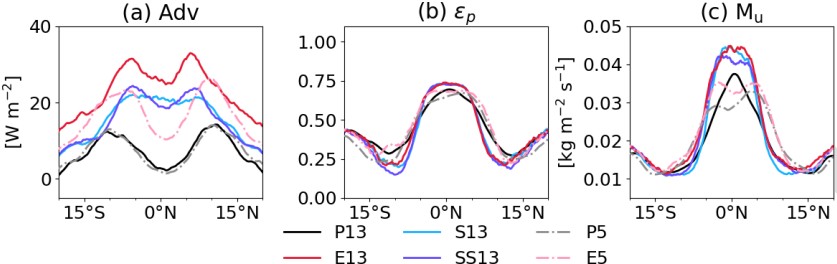

**Figure 10.** Time and zonal mean of **(a)** the meridional advection term ($d \rho V_h \cdot \nabla h$), **(b)** estimated precipitation efficiency, and **(c)** estimated convective mass flux using Eqs. ( 2) and (5).

term is similar between P13 and P5, it is overall lower for E5 than that for E13, particularly in the ITCZ where the advection term decreases almost by half (Table 3). Furthermore, the peaks are located further away in E5 than in E13. Despite these differences, the resulting changes in $\epsilon_p$ and $M_u$ are at best modest for E5 and E13 (Fig. 10b and c).

Similarly, SS13 shows large BL meridional advection in the ITCZ and has local maxima at around 7° N/S but overall weaker advection by around 6 W m$^{-2}$ than E13 (Fig. 10a). The advection consideration leads to a decrease in $M_u$ by 0.0013 kg m$^{-2}$ s$^{-1}$ and an increase in $\epsilon_p$ by 0.039 in the ITCZ (Table 3), which changes less strongly in the outer tropics (Fig. 10b and c). For S13 meridional advection closely follows that for SS13 (Fig. 10a), leading to averaged advection of 20.9 W m$^{-2}$ in the ITCZ (Table 3). However, the maximum value is located in the ITCZ rather than at around 7° N/S. This difference between S13 and SS13 is because the minor increase from SS13 to S13 in $\overline{U}_h$ in the ITCZ (Fig. 5a) and the slightly reduced meridional $h$ gradient for S13 (Fig. 7a and b). Despite this difference in shape, $M_u$ and $\epsilon_p$ in S13 and SS13 change by almost the same degree when considering meridional advection.

In summary, the meridional advection slightly increases $\epsilon_p$ and slightly decreases $M_u$ for all cases. For example, $\epsilon_p$ increases in the ITCZ by 8 % from P13 to the other coarse-resolution runs (E13, S13 and SS13), whereas it is almost identical without considering the advection term (Sect. 5.5). However, the main differences among the runs are evident in

$M_u$, compared to $\epsilon_p$ and $\langle q_v \rangle$, meaning that the close association between $M_u$ and rainfall remains strong.

## 6 Conclusions

Over decades, general circulation models have shown disagreement on tropical rainfall distributions, demonstrating a high level of uncertainty. Idealized modeling frameworks, such as aquaplanet simulations, showed a great sensitivity of tropical rainfall to various factors. This study presented a novel diagnostic tool to identify links between the processes important for rainfall in a fully coupled and physically consistent way. The innovation of our diagnostic tool is the application of the conceptual framework by Emanuel (2019) to output from a numerical model. Amongst other things, the framework assumes mass and energy conservation as well as the boundary-layer quasi-equilibrium (BLQE) (Raymond, 1995). BLQE describes the balance of moist static energy in the BL between surface enthalpy fluxes and vertical advection through convective downdrafts and large-scale subsidence.

We applied our diagnostic tool to tropical aquachannel experiments using the ICON-NWP model. The experiments vary with treatments of shallow and deep convection and with different horizontal grid spacings (13 and 5 km). The channel geometry is designed with a zonal extension as large as the Earth's circumference and a meridional extension between 30° N and 30° S where time-invariant, zonally constant variables are prescribed. The SSTs are prescribed with a

zonally symmetric distribution and maximum at the Equator (Neale and Hoskins, 2000).

All experiments show an ITCZ at the Equator and a Hadley circulation with an ascending branch at the Equator and descending branches at 15° N and 15° S – somewhat narrower than the Hadley circulation in reality – and accompanying easterly trade winds at the flanks of the ITCZ. The confinement of the Hadley circulation between 15° N and 15° S is because the model develops its own internal circulation, at least partly related to suppressed eddy fluxes at the rigid walls. Despite the similar structures among the experiments, there are differences due to changes in convective treatment, mainly deep convection, and in horizontal resolution. From parameterized to explicit deep convection, the maximum precipitation in the ITCZ increases and the Hadley circulation becomes stronger. These changes are more pronounced at 13 km than at 5 km, which shows reduced rainfall in the ITCZ. Figure 11 illustrates how variables are relevant to rainfall change in response to different model configurations. We summarize the changes focusing on the ITCZ region as follows.

– *Dependence on convective treatment at 13 km.* From parameterized (Fig. 11a) to explicit deep convection (Fig. 11b), the rainfall amount increases substantially and the large-scale circulation and surface horizontal winds get stronger. Strong surface winds enhance surface enthalpy fluxes by 20 %–28 %. The vertical difference in moist static energy between the BL and the lower troposphere is relatively robust to changing convective treatment. Somewhat surprisingly, precipitation efficiency is little sensitive to the representation of convection with values of 0.633–0.657. In contrast, convective updraft mass flux increases by 30 %–38 % with explicit deep convection. With the constant value of precipitation efficiency, convective updraft mass flux increases proportionally to increasing convective downdraft mass flux, which is balanced by enhanced surface enthalpy fluxes to maintain BLQE. Thus, the rainfall change in response to convective treatment at 13 km is due to the tight links among dynamical fields, surface fluxes and convective mass flux.

– *Dependence on resolution with the same convective treatment.* From the coarse (Fig. 11b) to high resolutions (Fig. 11c), the ITCZ becomes drier, which is slightly further into the Northern Hemisphere. The higher resolution alters the strength of the Hadley circulation and even more so surface winds, which are overall weakened except for the trade wind belt in P5 where winds get intensified. Weakened surface winds suppress surface enthalpy fluxes in the ITCZ. Additionally, the vertical contrast in moist static energy in the ITCZ becomes larger with increasing resolution. The combination of suppressed surface enthalpy fluxes and

increased vertical contrast in moist static energy are associated with reduced convective mass flux, while precipitation efficiency changes little due to increasing resolution. This underlines the importance of BLQE to understand rainfall difference due to changing resolution. Thus, both dynamics and thermodynamics become important to understand the sensitivity of mean rainfall to resolution.

– *Dependence on convective treatment at 5 km.* The sensitivity of rainfall to convective treatment at 5 km exhibits some differences from that at 13 km. As for the other cases, explicit convection produces more rainfall than parameterized convection. The difference in rainfall is associated primarily with differences in convective updraft mass flux related to BLQE, while precipitation efficiency remains largely unaffected by convective treatment. At 5 km, however, the differences in surface enthalpy fluxes are not due to differences in surface wind speed, which is relatively similar in the two runs, although the Hadley circulation is stronger in E5 than in P5. Instead, a larger moisture contrast between the ocean surface and the air in E5 enhances surface enthalpy fluxes and convective mass flux. Furthermore, the effects of stronger convective mass flux on rainfall are partially offset by relatively low column averaged humidity, of which the differences in moisture in the lower troposphere contribute the most. Thus, BLQE is still key to understanding the dependence on convective treatment at 5 km, yet the balance is achieved by thermodynamics within and above the BL, while dynamic fields are less involved.

The model configuration changes radiative cooling and dry stability in all latitudes, but these changes compensate for each other, having a very small net effect on convective mass flux. Note that radiative cooling was found to be crucial for radiative convective equilibrium without a large-scale circulation (Emanuel, 2019), but this is not the case for our experiments with full physics and dynamics. Moreover, explicit deep convection can produce more delicate distributions of convection, such as deep, shallow and congestus clouds, than parameterized convection, but again mean rainfall is insensitive to a change in radiative cooling associated with these structures. With explicit deep convection, the meridional advection of moist static energy in the BL is not negligible, leading to a slight increase in precipitation efficiency of 0.03–0.05. However, convective updraft mass flux still exhibits the strongest association with rainfall. A caveat of this diagnostic tool is that the effects of entrainment and detrainment are not considered, which might be important for convective updraft mass flux (Zipser, 2003; Möbis and Stevens, 2012). Somewhat indirectly, these effects are included in $h_b - h_m$ through lower-tropospheric $h$ and in $\epsilon_p$ through indirect effects of subcloud evaporation. Furthermore, the role of thermodynamics in the lower troposphere

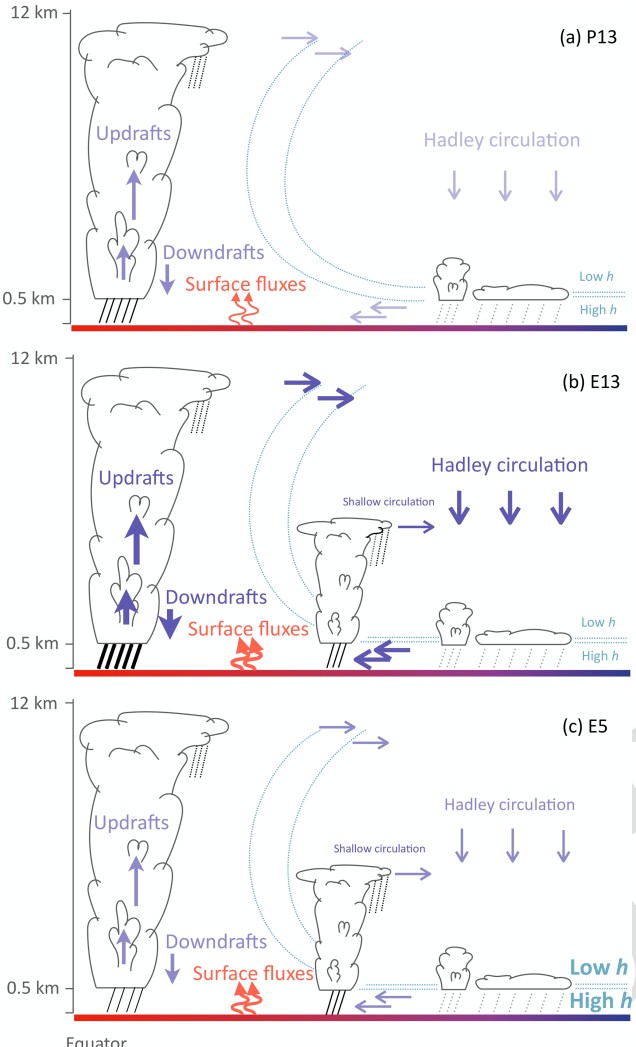

**Figure 11.** Schematic depiction of the important processes associated with rainfall for **(a)** P13, **(b)** E13 and **(c)** E5. The thick (thin) arrows and lines indicate large (small) quantities. The large fonts indicate processes important to understand the mean rainfall. Precipitation efficiency is not shown as it hardly changes due to model configuration. The dotted curves indicate contour lines for constant values of moist static energy. See Sect. 6 for a detailed discussion.

may become more important when using a slab ocean model (Tompkins and Semie, 2021) or different turbulence and/or microphysics schemes (Lang et al., 2023).

The merit of our diagnostic tool lies in a fair comparison of simulations with different representations of convection to examine the processes potentially linked to rainfall. Since those processes are strongly coupled to each other, it is not trivial to disentangle what processes are ultimately responsible for rainfall. Furthermore, explicit and parameterized convection treats vertical motion differently, so it is inconsistent to compare convective updraft mass flux obtained directly from the modeled vertical wind field. Thus, we emphasize

that the diagnostic tool presented here provides a physically consistent, fair comparison between explicit and parameterized convection and helps obtain a quantitative and qualitative view on important links in the system. Although the conclusion of this study may not hold using other simulation geometry such as aquaplanet, the application of the ITCZ diagnostic tool will help gain a deeper understanding of processes responsible for mean rainfall distribution. Lastly, this tool also has potential to specify sources of uncertainty in NWP models and to identify the reasons behind the large spread in ITCZ behavior among different climate models.

## Appendix A: Emanuel's (2019) framework

In the framework of Emanuel (2019), the large-scale vertical velocity at the top of the boundary layer (BL) $w$ is written as

$$\rho w = M_{\mathrm{u}} - M_{\mathrm{d}} - \rho w_{\mathrm{e}}, \tag{A1}$$

where $M_{\mathrm{u}}$ and $M_{\mathrm{d}}$ are convective upward and downward mass fluxes of water vapor in $\mathrm{kg\,m^{-2}\,s^{-1}}$, respectively; $w_{\mathrm{e}}$ is the environmental vertical velocity away from convection; and $\rho$ is the air density at the top of the BL. Note that Emanuel (2019) uses dimensionless mass flux and vertical velocity fields, but we prefer utilizing them in physical units in order to apply the conceptual model to the simulated fields. Microphysical processes are not treated explicitly but formulated through one constant parameter, the so-called precipitation efficiency $\epsilon_{\mathrm{p}}$, which represents the fraction of all condensate that reaches the ground as precipitation. Also, $\epsilon_{\mathrm{p}}$ can then be used to parameterize $M_{\mathrm{d}}$ as a function of $M_{\mathrm{u}}$ in the following way: $M_{\mathrm{d}} = (1 - \epsilon_{\mathrm{p}})M_{\mathrm{u}}$.

With conservation of moist static energy, $h$ budget for the BL becomes

$$\int_b \left( \rho \frac{\partial h}{\partial t} + \rho \boldsymbol{V} \cdot \nabla h \right) \mathrm{d}z = F_{\mathrm{h}} - \int_b \dot{Q} \mathrm{d}z, \tag{A2}$$

where $\boldsymbol{V}$ is the three-dimensional wind velocity, $F_{\mathrm{h}}$ is the surface enthalpy flux, $\dot{Q}$ is the radiative cooling and the subscript $b$ indicates the integral over the depth of the BL. In a well-mixed BL the vertical advection of $h$ occurs at the top of the BL, and boundary-layer quasi-equilibrium (BLQE) assumes that the injection of low-$h$ air by convective downdrafts ($M_{\mathrm{d}}$) and large-scale subsidence ($w_{\mathrm{e}}$) is balanced by the uptake of high $h$ through surface fluxes (Raymond, 1995). Therefore, the vertical advection can be represented by a simple difference between characteristic values of $h$ for the BL ($h_{\mathrm{b}}$) and the free troposphere ($h_{\mathrm{m}}$), here denoted by $h_{\mathrm{b}} - h_{\mathrm{m}}$. In quasi-equilibrium, the local time derivative vanishes and Eq. A2 becomes

$$\mathrm{d}\rho \boldsymbol{V}_{\mathrm{h}} \cdot \nabla h = F_{\mathrm{h}} - \mathrm{d}\dot{Q}_{\mathrm{b}} - (M_{\mathrm{d}} + \rho w_{\mathrm{e}})(h_{\mathrm{b}} - h_{\mathrm{m}}), \tag{A3}$$

where d is the BL height, $\mathbf{V}_\mathrm{h}$ is the horizontal wind velocity and $\dot{Q}_\mathrm{b}$ is the radiative cooling at the top of the BL, which is assumed to be characteristic for the entire BL, i.e., constant. In addition, advection is assumed to be approximately constant throughout the BL. Assuming that d is small, net radiative cooling at the top of the BL and the horizontal advection of $h$ will be small and can be neglected. Then, Eq. (A3) TS5 becomes

$$0 = F_\mathrm{h} - (M_\mathrm{d} + \rho w_\mathrm{e})(h_\mathrm{b} - h_\mathrm{m}). \tag{A4}$$

The weak temperature gradient approximation implies that horizontal advection in the thermodynamic equation can be neglected, and time changes also vanish in quasi-equilibrium or steady state, such that thermodynamic balance is between vertical advection and diabatic heating (Sobel et al., 2001). In an ascending region, condensational heating is balanced by adiabatic cooling by an ascending parcel. In a descending region, of which the area fraction is far larger than an ascending region, adiabatic warming by subsidence is balanced by radiative cooling. The thermodynamic balance in the descending region can be formulated as $\rho w_\mathrm{mid} S = \dot{Q}$, where $w_\mathrm{mid}$ is the descending motion in the free troposphere and $S \equiv c_p \frac{\mathrm{d}T}{\mathrm{d}z} + g$ is closely related to dry static stability, with $g$ the gravitational acceleration. Assuming mass conservation and approximately constant vertical velocity, $w_\mathrm{e}$ and $w_\mathrm{mid}$ are approximated to be identical. Thus, using the thermodynamic balance, Eq. (A1) can be further written as

$$(\epsilon_\mathrm{p} M_\mathrm{u} - \rho w) = \frac{\dot{Q}}{S}, \tag{A5}$$

which illustrates the limitation of convection by longwave cooling in the environment.

Using Eqs. (A1), (A4) and (A5), we can then derive a diagnostic expression for $M_\mathrm{u}$ as

$$M_\mathrm{u} = \frac{1}{1 - \epsilon_\mathrm{p}} \left( \frac{F_\mathrm{h}}{h_\mathrm{b} - h_\mathrm{m}} - \frac{\dot{Q}}{S} \right). \tag{A6}$$

The above formulation of $M_\mathrm{u}$ is employed by the ITCZ diagnostic tool presented in Sect. 4.

*Code availability.* The diagnostic tools can be found at: https://doi.org/10.5281/zenodo.10220388 (Jung et al., 2023 TS6).

*Data availability.* Model output is published on Open Data LMU – Physics (https://doi.org/10.57970/P3AHB-YBA70) and is available for download (Ruckstuhl et al., 2023).

*Supplement.* The supplement related to this article is available online at: https://doi.org/10.5194/wcd-5-1-2023-supplement.

*Video supplement.* This paper includes a video supplement, which shows a series of snapshots of precipitable water (shading) and rainfall rate (contour) in the tropical aquachannel simulations. The beginning of the analysis period is referred to as day 0, corresponding to the solid vertical lines in Fig. 1. TS7

*Author contributions.* The project for the aquachannel runs was initially proposed by PK, CH and TJ. YR and RR set up and performed the simulations with advice from PK, CH and TJ. HJ constructed the diagnostic tool with advice from PK. HJ conducted the analyses and visualized them. HJ and PK interpreted the results. HJ wrote the first version of the manuscript. All authors reviewed and edited it.

*Competing interests.* At least one of the (co-)authors is a member of the editorial board of *Weather and Climate Dynamics*. The peer-review process was guided by an independent editor, and the authors also have no other competing interests to declare.

ther geographical representation in this paper. While Copernicus Publications makes every effort to include appropriate place names, the final responsibility lies with the authors.

*Acknowledgements.* Hyunju Jung thanks Kerry Emanuel for the email exchange that initiated the use of his framework and Ron McTaggart-Cowan for suggesting the surface flux diagnostic framework. We also thank Oriol Tinto Prims for resolving technical issues regarding the simulations and three anonymous reviewers for their thoughtful comments that improved our paper. The research leading to these results has been performed within the subproject "B6" of the Transregional Collaborative Research Center SF-B/TRR 165 "Waves to Weather" (Craig et al., 2021), http://www.wavestoweather.de (last access: 29 November 2023), funded by the German Research Foundation (DFG).

*Financial support.* This research has been supported by the Deutsche Forschungsgemeinschaft (grant no. SFB/TRR 165).

*Review statement.* This paper was edited by Martin Singh and reviewed by three anonymous referees.

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

**Remarks from the language copy-editor**

CE1  Please note the slight adjustment here.

**Remarks from the typesetter**

TS1  Please give an explanation of why this needs to be changed. We have to ask the handling editor for approval. Please provide a \*.pdf and mark the instances that need to be changed with an explanation. Thank you.

TS2  Please give an explanation of why this needs to be changed. We have to ask the handling editor for approval. Please provide a \*.pdf and mark the instances that need to be changed with an explanation. Thank you.

TS3  Please give an explanation of why this needs to be changed. We have to ask the handling editor for approval. Please provide a \*.pdf and mark the instances that need to be changed with an explanation. Thank you.

TS4  Please confirm.

TS5  It is our house standard to add parentheses around equation numbers within the text. We only delete them when they are mentioned within additional parentheses. Thank you for your understanding.

TS6  Please confirm.

TS7  Please upload this video supplement to a repository as we need a link for this section. For more information, please refer to https://www.weather-climate-dynamics.net/submission.html.

TS8  Please confirm reference list entry.