# Peer review of "Understanding the dependence of mean precipitation on convective treatment and horizontal resolution in tropical aquachannel experiments"

_Weather and Climate Dynamics, 2023_

## Author Comment (AC2)

Response to comments of Referee #2

We thank the reviewer for the thoughtful and constructive comments on our manuscript. We have been carefully considering each of the comments. The reviewer's comments are repeated in normal font and our responses are followed in blue.

The novel contribution of this paper is to use the diagnostic framework of Emanuel (2019) to help understand the physical reasons for the differences among a set of 4 experiments using an aquachannel model with prescribed, zonally symmetric sea surface temperature. The 4 experiments differ only in whether or not they use a parameterization of deep convection or one of two parameterizations of shallow convection. The horizontal grid spacing is 13 km, so deep convection is poorly resolved and shallow convection is not really resolved at all. Even so, understanding why the results differ, even if the results are seriously compromised relative to nature, is a step forward, so I am in favor of seeing some version of the paper published with this strong caveat.

As both reviewers pointed out the caveat regarding the model resolution, we plan to include 5-km aquachannel simulations with explicit and parameterized deep and shallow convection. The configuration of them corresponds to E13 and P13, except for horizontal resolution. The high-res simulations show that mean tropical rainfall depends on resolution (Fig. 1). We apply the ITCZ diagnostics presented in the submitted manuscript to investigate what processes are responsible for the sensitivities. There are some changes due to the resolution dependency, such that the vertical difference in moist static energy ($h_b$-$h_m$) becomes important for rainfall differences while surface enthalpy fluxes still play a crucial role. We plan to include the 5-km simulations and results in the revised manuscript by describing sensitivities of mean tropical rainfall to convective treatment and model resolution.

[Figure]

Figure 1. Distributions of (a) time and zonal mean of precipitation rate and (b) precipitation intensity between 20°N/S. Precipitation fields are coarsened on a 0.2° lat-lon grid using a conservative remapping. P5 is the 5-km aquachannel simulation with parameterized deep and shallow convection and E5 is the one with explicit deep and shallow convection.

The Emanuel framework consists of diagnostic equations for cumulus updraft mass flux, large-scale vertical motion, and a single predictive equation for the mass-weighted vertically integral of the moist static energy. In the original paper, it was used as a tool for very basic understanding of tropical circulations. Here it is being used instead to help diagnose and understand complex simulations, albeit in a simple aquachannel framework with steady, zonally symmetric SSTs.

Of the three equations in the original framework, the current authors use only one. It would be useful if they could explain why they chose only a single diagnostic. The most important criticism I have is that it is not made clear what is being specified and what is being calculated from this framework. I gather from a mediated, anonymous exchange with the authors that the models' precipitation, surface heat fluxes, radiative cooling, dry static stability, and difference between boundary layer and lower tropospheric moist static energy are being fed into the framework, and precipitation efficiency and updraft mass flux are being diagnosed. Whatever the case, the inputs and output(s) must be clearly stated. The sentence "In our diagnostics, $M_u$ and $epsilon_p$ are not obtained directly from vertical motion but indirectly using other consistent quantities" is far too vague. Perhaps just state that these quantities are diagnosed using (1) and (2) with inputs from the simulations.

Amongst the three equations in Emanuel (2019), we only picked the formulation of $M_u$ because this variable is directly related to precipitation through Eq. 2 in the submitted manuscript.

We thank the reviewer for spending time sorting out which variables are diagnosed through the framework and which ones are computed from model output. We will elaborate on this, clearly explaining why we chose the formulation of $M_u$ and what quantities are diagnosed in Sect. 4 where we describe the ITCZ diagnostics in the revised manuscript.

One clear difference among the simulations is that the parameterization of deep convection tends to weaken the Hadley circulation. The diagnostic framework does not really help us understand why. Since the output is precipitation efficiency and mass flux and everything else is fed in from the simulations themselves, one would suppose that the focus would be in the predicted quantities. To imply that the Hadley circulations in the simulations with no parameterization of deep convection are stronger because the wind-driven fluxes are stronger seems tautological. When the stronger fluxes are fed into the framework, it dutifully diagnoses a stronger convective mass flux in the ITCZ; not sure what we have learned. I think the authors are up against the age-old problem of inferring causality in a steady system. One might also point out that the specification of SST means that surface energy balance is not enforced; if a slab ocean were coupled it would not be able to sustain the large differences in turbulent heat fluxes observed among the experiments.

We would like to highlight that our focus is on mean rainfall and we mainly address "links" between processes important for rainfall. The links do not mean a unidirectional but multidirectional interaction. This means that we cannot disentangle if the increased rainfall

with explicit deep convection drives the stronger large-scale circulation, which can lead to enhanced surface fluxes, or if the stronger large-scale circulation results in the increased rainfall. What we identify is that the differences in rainfall and large-scale circulation are strongly coupled through boundary-layer quasi-equilibrium. The interesting point that the reviewer raised regarding the slab ocean model makes us wonder if the link between rainfall and large-scale circulation would get weaker, if we included atmosphere-ocean effects. We presume that the link may get weaker but cannot give a definite answer. We, however, believe that in this case the application of the ITCZ diagnostics presented in the submitted manuscript would certainly help find what processes are important! In the revised manuscript, we will clarify what problem we are addressing and will discuss about the limitation of the simulation setup associated with the prescribed SSTs.

One result that is fascinating is the constancy, across experiments, of the precipitation efficiency in the ITCZ region. It would be great if the authors could address this result.

We will describe and discuss precipitation efficiency in more detail in Sect. 5.5 in the revised manuscript.

Another improvement that very much help with the understanding of the diagnostic is to plot, either as part of Figure 4 or as a separate figure, the actual terms in (1); namely, the ratio of the surface heat flux to the moist static energy difference, and the ratio of the radiative cooling to the dry static stability.

We will add an additional plot to show the latitudinal distributions of the terms, $F_h/(h_b-h_m)$ and $Q/S$ in the revised manuscript.

One other question I have is why the authors chose the diagnostic equation for the cumulus updraft mass flux rather than the one for the large-scale vertical velocity. Is it because the latter is difficult to sample in the simulations? More difficult than sampling rainfall?

This is again because $M_u$ is directly linked to rainfall through Eq. 2. We will clarify this in the revised manuscript.

A few specific points:

Figure 1: As the authors note, the model does not seem to have settled down into a steady state by the end of the integration. It might be worth it to extend one of the 4 simulations beyond this ending time.

That is a good idea. Unfortunately, the project this work is embedded in is now coming to an end and we need to wrap up. In addition, the same aquachannel simulations have been used for a companion project for data assimilation (DA). As understanding meteorological information in the same simulations is critical for the DA project, we focus on the 40-day period. We will keep this aspect in mind for future planning.

Line 199: "We speculate that extreme rainfalls...."

This will be changed in the revised version.

Line 210-211: It is not necessarily true that the steady state must be equatorially symmetric. There can be spontaneous symmetry breaking.

We agree with that. We will address this point in the revised version

Equation 2: I would have thought that the water vapor concentration that appears here should be evaluated at cloud base rather that taking a vertical average.

We will check if the results would change using the water vapor concentration at cloud base and discuss this in a revised version. Given the fact that profiles of moist static energy differ mostly in the lower troposphere, implying differences in humidity, we expect the results not to change much.

Section 5.1.1: I understand the breakdown between wind and delta enthalpy, but why is it important to distinguish sensible from latent fluxes here?

Our intention is to show important contributing factors to mean $F_h$, the sum of surface sensible and latent heat fluxes. The surface sensible and latent fluxes share surface wind speed, but thermodynamic variables ($\Delta q$ and $\Delta T$) are different. So, it is interesting to examine if the thermodynamic conditions also differ and whether it is mainly due to temperature or moisture. For example, convective downdrafts transport colder and moister air into the boundary layer, which can lead to substantial influences of thermodynamics on surface fluxes. It is not our case, but it may be if one uses a slab ocean model (Tompkins and Semie, 2021), as the reviewer mentioned. Therefore, it is worth exploring and demonstrating the different thermodynamic influences separately. In the revised manuscript, we will clarify why we distinguish surface latent and sensible heat fluxes in the context of Sect. 5.1.1.

Line 633-634: If radiative cooling is shut off, there can be no latent heating that, over the whole domain, must balance the cooling. The system would shut down.

What we mean is a change or perturbation of radiative cooling. It does not mean that there is no radiative cooling at all. We will clarify this in the revised manuscript.

Reference
Tompkins, A. M., & Semie, A. G. (2021). Impact of a mixed ocean layer and the diurnal cycle on convective aggregation. *Journal of Advances in Modeling Earth Systems*, 13, e2020MS002186. https://doi.org/10.1029/2020MS002186

---

## Author Response (AR1)

Response to comments of Referee #1

We thank the reviewer for the thoughtful and constructive comments on the manuscript. We have been carefully considering each of the comments. The reviewer's comments are repeated in normal font and our responses are followed in blue.

This study investigates the impact of convective treatment (particularly, parameterized vs explicit deep convection) on the simulation of mean tropical precipitation (particularly, the ITCZ), using a 13km ICON model in a semi-aquaplanet simulation with walls at 30N/S. They find, with explicit deep convection, rainfall in the ITCZ increases by 35% and the Hadley circulation as well as surface winds become stronger. Based on a diagnostic framework based on Emanuel (2019), they attribute the difference to the stronger surface horizontal winds with explicit deep convection, which modifies the boundary layer equilibrium and consequently the updraft mass flux.

I have some concerns about the model setup and the derivation of Eq. 1. I suggest a major revision with the following comments.

Major comments:

1. Uncertainty due to model setup

Resolution:
The effect of explicit versus parameterized deep convection is investigated at a horizontal resolution of 13 km. It is generally believed that the horizontal resolution needed to partially resolve deep convection should be ~1km, the 13km resolution used here is not sufficient to resolve deep convection, so the setup of the S13 experiment would not be recommended. It is unclear how sensitivity is the effect of explicit deep convection to the background horizontal resolution. Will the conclusion be different if a higher horizontal resolution, e.g. 3km, is used?

Thank you to raise this important point. We originally planned to include higher-resolution simulations with a grid spacing of 5 km, but then had to abandon this for technical reasons. Fortunately, the simulations have now become available as originally planned. Their convective treatments correspond to those of E13 and P13 in the submitted manuscript. This entire set of aquachannel simulations exhibits that mean tropical rainfall is sensitive to convective treatment and horizontal resolution (Fig. 1). We apply the ITCZ diagnostics presented in the submitted manuscript to investigate what processes are responsible for the sensitivities. With increasing resolution, the vertical difference in moist static energy ($h_b$-$h_m$) gains importance to shape mean rainfall while the role of surface enthalpy fluxes remains substantial. The manuscript has been updated including the 5-km runs. Please note that quantities, e.g., in tables 2 and 3, have changed due to conservative remapping to compare the 5- and 13-km runs.

[Figure]

*Figure 1. Distributions of (a) time and zonal mean of precipitation rate and (b) precipitation intensity between 20°N/S. Precipitation fields are coarsened on a 0.2° lat-lon grid using a conservative remapping. P5 is the 5-km aquachannel simulation with parameterized deep and shallow convection and E5 is the one with explicit deep and shallow convection.*

Walls at 30N/S:
The aquaplanet simulations has a wall at 30N/S. This setup is likely to affect many aspects of the simulations including the ITCZ. It is unclear to me if the conclusion of this study would be different if there is no wall but a global aquaplanet.

We agree that the closed walls at 30N/S exclude many important aspects such as tropical-extratropical interactions on the ITCZ and leads to a narrower-than-normal Hadley circulation. On the other hand, this exclusion simplifies our problem, such that we can concentrate on processes in the equatorial region in full isolation, which we deem an important building block in understanding predictability in the tropics fully in subsequent works. The important role of surface enthalpy fluxes for mean rainfall may not hold to the same degree using an aquaplanet simulation, but we emphasize that the ITCZ diagnostics we propose in our manuscript can, of course, be applied to any simulation geometry to test the sensitivity of mean rainfall. So, we see this also as a first step towards a more comprehensive intercomparison in the long run, which will help gain a deeper understanding of important processes. We include this aspect of using different geometry in Sect. 2.2 and will leave the sensitivity of tropical rainfall to simulation geometry for future study.

L168-169: "We presume that a wider channel, a two-way nested channel within a global domain or an aquaplanet would simulate jets at a more realistic location and may affect many aspects, particularly associated with tropical-extratropical interactions. However, the channel geometry suppresses these interactions and thus reduces complexity."

L732-733: "Although the conclusion of this study may not hold using other simulation geometry such as aquaplanet, the application of the ITCZ diagnostic will help gain a deeper understanding of processes responsible for mean rainfall distribution."

2, The derivation of Eq. A5, which leads to Eq. 1

Eq. A1 is for the top of the BL (the subsidence is w_e) while the equation at L686 is from the balance between radiative cooling and descending is for free troposphere (i.e., the subsidence is not w_e), then, how could these two equations be combined into Eq. A5.

Right, they are not identical. Assuming mass conservation and approximately constant vertical velocity, $w_e$ at the top of BL and in the free troposphere are similar, which was also argued in Emanuel (2019) for simplification. Besides, we believe that the beauty of his framework lies in traceability, providing the simplest solution that one could come up with to explain atmospheric phenomena in the tropics. What we learned from our research is that despite its simplicity, it is nontrivial to understand the behavior of mean tropical rainfall. We have elaborated explaining different $w_e$ in Appendix A in a revised manuscript.

L768-771: "The thermodynamic balance in the descending region can be formulated as $\rho\, w_{\mathrm{mid}}\, S = \dot{Q}$ is the descending motion in the free troposphere and $S \equiv c_p \frac{dT}{dz} + g$ is closely related to dry static stability with $g$ the gravitational acceleration. Assuming mass conservation and approximately constant vertical velocity, $w_e$ and $w_{\mathrm{mid}}$ are approximated to be identical."

Minor comments:

L54: Not sure if "appropriate" is the right word here. Each model center has chosen the model resolution appropriately, according to their needs and computational resources.

Thank you for the insightful thought. We have changed the wording in the revised manuscript.

L55-57: "Given specific purposes and computational resources, a horizontal grid spacing of < 10km can be selected to resolve deep convection (Weisman et al., 1997; Hong and Dudhia, 2012; Prein et al., 2015) with some extreme limit of 100 m  (Kwon and Hong, 2017; Jeevanjee, 2017)."

L60-70: According to Zhou et al. (2022), the storm-resolving simulation (res ~3km) does not reduce the bias in tropical precipitation characteristics (except for the better representation of strong convection events and tropical cyclones) and is not likely to alleviate the double-ITCZ bias.

Zhou W., L.R. Leung, J. Lu, (2022): Linking large-scale double-ITCZ bias to local-scale drizzling bias in climate models. Journal of Climate 35 (24), 4365-4379.

We have included the suggested paper when we have discussed add values of convection-permitting models in the revised manuscript.

L67-68: "Despite these many improvements, convection-permitting models do not always guarantee alleviating the long-standing ITCZ problem (Zhou et al., 2022)"

L71: resolving (deep) convection

This have changed in the revised manuscript.

I suggest moving section 4 (description of the diagnostic framework) to section 2.

We appreciate the suggestion and discussed it at length. At the end, we decided to leave the structure largely as it is. The diagnostic framework is in some sense part of results while being a method at the same time. Placing the description of the method closer to the application certainly has the advantage for the reader to better remember the meaning of the individual terms and their relationships.  To avoid confusion or creating false expectations, we have decided to rename Sect. 2 as "Aquachannel experiments".

Response to comments of Referee #2

We thank the reviewer for the thoughtful and constructive comments on our manuscript. We have been carefully considering each of the comments. The reviewer's comments are repeated in normal font and our responses are followed in blue.

The novel contribution of this paper is to use the diagnostic framework of Emanuel (2019) to help understand the physical reasons for the differences among a set of 4 experiments using an aquachannel model with prescribed, zonally symmetric sea surface temperature. The 4 experiments differ only in whether or not they use a parameterization of deep convection or one of two parameterizations of shallow convection. The horizontal grid spacing is 13 km, so deep convection is poorly resolved and shallow convection is not really resolved at all. Even so, understanding why the results differ, even if the results are seriously compromised relative to nature, is a step forward, so I am in favor of seeing some version of the paper published with this strong caveat.

As both reviewers pointed out the caveat regarding the model resolution, we have included 5-km aquachannel simulations with explicit and parameterized deep and shallow convection in the revised manuscript. The configuration of them corresponds to E13 and P13, except for horizontal resolution. The high-res simulations show that mean tropical rainfall depends on resolution (Fig. 1). We apply the ITCZ diagnostics presented in the submitted manuscript to investigate what processes are responsible for the sensitivities. There are some changes due to the resolution dependency, such that the vertical difference in moist static energy ($h_b$-$h_m$) becomes important for rainfall differences while surface enthalpy fluxes still play a crucial role. The manuscript has been updated including the 5-km runs. Please note that quantities, e.g., in tables 2 and 3, have changed due to conservative remapping to compare the 5- and 13-km runs.

[Figure]

Figure 1. Distributions of (a) time and zonal mean of precipitation rate and (b) precipitation intensity between 20°N/S. Precipitation fields are coarsened on a 0.2° lat-lon grid using a conservative remapping. P5 is the 5-km aquachannel simulation with parameterized deep and shallow convection and E5 is the one with explicit deep and shallow convection.

The Emanuel framework consists of diagnostic equations for cumulus updraft mass flux, large-scale vertical motion, and a single predictive equation for the mass-weighted vertically integral of the moist static energy. In the original paper, it was used as a tool for very basic understanding of tropical circulations. Here it is being used instead to help diagnose and understand complex simulations, albeit in a simple aquachannel framework with steady, zonally symmetric SSTs.

Of the three equations in the original framework, the current authors use only one. It would be useful if they could explain why they chose only a single diagnostic. The most important criticism I have is that it is not made clear what is being specified and what is being calculated from this framework. I gather from a mediated, anonymous exchange with the authors that the models' precipitation, surface heat fluxes, radiative cooling, dry static stability, and difference between boundary layer and lower tropospheric moist static energy are being fed into the framework, and precipitation efficiency and updraft mass flux are being diagnosed. Whatever the case, the inputs and output(s) must be clearly stated. The sentence "In our diagnostics, M_u and epsilon_p are not obtained directly from vertical motion but indirectly using other consistent quantities" is far too vague. Perhaps just state that these quantities are diagnosed using (1) and (2) with inputs from the simulations.

Amongst the three equations in Emanuel (2019), we only picked the formulation of $M_u$ because this variable is directly related to precipitation through Eq. 2 in the submitted manuscript.

We thank the reviewer for spending time sorting out which variables are diagnosed through the framework and which ones are computed from model output. We have elaborated on this, clearly explaining why we chose the formulation of $M_u$ and what quantities are diagnosed in Sect. 4 where we describe the ITCZ diagnostics in the revised manuscript.

L277-279: "Amongst the three equations in the original framework, we only use the formulation of convective updraft mass flux, which can be directly related to precipitation. We refer to Emanuel (2019) for the complete derivation of the conceptual framework."

L319: "In other words, $F_h$, $h_b - h_m$, $\dot{Q}$, $S$, Pr and $\langle q_v \rangle$ are fed into the two independent equations (1 and 2) to estimate $M_u$ and $\epsilon_p$."

One clear difference among the simulations is that the parameterization of deep convection tends to weaken the Hadley circulation. The diagnostic framework does not really help us understand why. Since the output is precipitation efficiency and mass flux and everything else is fed in from the simulations themselves, one would suppose that the focus would be in the predicted quantities. To imply that the Hadley circulations in the simulations with no parameterization of deep convection are stronger because the wind-driven fluxes are stronger seems tautological. When the stronger fluxes are fed into the framework, it dutifully diagnoses a stronger convective mass flux in the ITCZ; not sure what we have learned. I think the authors are up against the age-old problem of inferring causality in a steady system. One might also

point out that the specification of SST means that surface energy balance is not enforced; if a slab ocean were coupled it would not be able to sustain the large differences in turbulent heat fluxes observed among the experiments.

We would like to highlight that our focus is on mean rainfall and we mainly address "links" between processes important for rainfall. The links do not mean a unidirectional but multidirectional interaction. This means that we cannot disentangle if the increased rainfall with explicit deep convection drives the stronger large-scale circulation, which can lead to enhanced surface fluxes, or if the stronger large-scale circulation results in the increased rainfall. What we identify is that the differences in rainfall and large-scale circulation are strongly coupled through boundary-layer quasi-equilibrium. The interesting point that the reviewer raised regarding the slab ocean model makes us wonder if the link between rainfall and large-scale circulation would get weaker, if we included atmosphere-ocean effects. We presume that the link may get weaker but cannot give a definite answer. We, however, believe that in this case the application of the ITCZ diagnostics presented in the submitted manuscript would certainly help find what processes are important! In the revised manuscript, we have clarified what problem we are addressing and have discussed about the limitation of the simulation setup associated with the prescribed SSTs.

L617-618: "Note that these links are not unidirectional but multidirectional interactions in the sense that we cannot disentangle whether a stronger circulation leads to more rainfall or vice versa."

L666: "This study presented a novel diagnostic tool to identify links between the processes important for rainfall in a fully coupled and physically consistent way."

L723-725: "Furthermore, the role of thermodynamics in the lower troposphere may become more important when using a slab ocean model (Tompkins and Semie, 2021) or different turbulence and/or microphysics schemes (Lang et al., 2023}."

One result that is fascinating is the constancy, across experiments, of the precipitation efficiency in the ITCZ region. It would be great if the authors could address this result.

We have described and discussed precipitation efficiency in more detail in Sect. 5.5 in the revised manuscript.

L589-592:" Surprisingly, in all experiments $\epsilon_p$ has the maximum there with very similar values of 0.63-0.657 (Fig.4h and Table.3). Note that the time-averaged quantities are taken into account here, but timely varying Pr and $\epsilon_p$ can be strongly correlated (Narsey et al., 2019; Muller and Takayabu, 2020}. Furthermore, $\epsilon_p$ can depend significantly on how convection is treated in models (Li et al., 2022), but the different convective treatments do not alter $\epsilon_p$ in the ITCZ in our case."

Another improvement that very much help with the understanding of the diagnostic is to plot, either as part of Figure 4 or as a separate figure, the actual terms in (1); namely, the ratio of the surface heat flux to the moist static energy difference, and the ratio of the radiative cooling to the dry static stability.

Thank you for the suggestion. We have added the latitudinal distributions of the terms, $F_h/(h_b-h_m)$ and $\dot{Q}/S$ in Fig. 4 in the revised manuscript.

[Figure]

Figure 2. Time and zonal mean of (a) the surface enthalpy flux, (b) the vertical difference in moist static energy, (c) the ratio of the surface enthalpy flux and the vertical difference in moist static energy, (d) the lower tropospheric radiative cooling, (e) the dry static stability, (f) the ratio of the lower tropospheric radiative cooling and dry static stability, (g) estimated convective mass flux, (h) estimated precipitation efficiency, and (i) the column averaged specific humidity. Ranges of the y-axes in (c) and (f) are identical to facilitate comparison.

One other question I have is why the authors chose the diagnostic equation for the cumulus updraft mass flux rather than the one for the large-scale vertical velocity. Is it because the latter is difficult to sample in the simulations? More difficult than sampling rainfall?

This is again because $M_u$ is directly linked to rainfall through Eq. 2. We have clarified this in the revised manuscript.

L277-279: "Amongst the three equations in the original framework, we only use the formulation of convective updraft mass flux, which can be directly related to precipitation."

A few specific points:

Figure 1: As the authors note, the model does not seem to have settled down into a steady state by the end of the integration. It might be worth it to extend one of the 4 simulations beyond this ending time.

That is a good idea. Unfortunately, the project this work is embedded in is now coming to an end and we need to wrap up. In addition, the same aquachannel simulations have been used for a companion project for data assimilation (DA). As understanding meteorological information in the same simulations is critical for the DA project, we focus on the 40-day period. We will keep this aspect in mind for future planning.

Line 199: "We speculate that extreme rainfalls…."

Thank you for the input, but we have entirely reformulated here to include the effects of resolution.

L226-229: "To initiate deep convection explicitly, the model needs to develop instability on a grid box scale. The larger the grid box (or the coarser the grid resolution), the more instability can be accumulated over time, which in turn produces more intense rainfall (Weisman et al., 1997) and occasionally intense gridpoint storms (Giorgi, 1991; Scinocca and McFarlane, 2004)."

Line 210-211: It is not necessarily true that the steady state must be equatorially symmetric. There can be spontaneous symmetry breaking.

We agree with that. We have addressed this point in the revised version.

L238-240: "The remaining small asymmetries, which occur despite the symmetric nature of our simulation setup, are a further indication that the simulations may not have fully reached equilibrium or that there can be spontaneous symmetry breaking through internal variability."

Equation 2: I would have thought that the water vapor concentration that appears here should be evaluated at cloud base rather that taking a vertical average.

Figure 3g-i show the results including the water vapor concentration at the cloud base. The quantities in Fig. 3i (water vapor concentration at the cloud base) are greater than those in Fig. 2i (column averaged specific humidity weighted by column density) and thus the magnitudes in $M_u$ and $\epsilon_p$ get smaller in Fig. 3g and h, compared to those in Fig. 2g and h. Also, there are some differences in $M_u$ in the trade wind belts, compared to Fig. 2g, but rainfall differences are small there. Despite that, overall structures (differences between the runs) are robust. The selection of moisture field does not influence our conclusions but only scaling. Furthermore, we also tested it using the average specific humidity in the subcloud layer, but the results are very similar. We have included this sensitivity test for the choice of the moisture field in Eq. 2 in the revised manuscript.

L310-311: Precipitation can be related to the water vapor concentration at the subcloud layer or the average specific humidity of the subcloud layer rather than $\langle q_v \rangle$. We tested different choices of the thermodynamic variable in Eq.2, but it does not influence our results but only scaling.

[Figure]

Figure 3. As in Fig. 2 but (i) specific humidity at the cloud base, and (g) convective updraft mass flux and (h) precipitation efficiency using the specific humidity at the cloud base instead of column averaged humidity.

Section 5.1.1: I understand the breakdown between wind and delta enthalpy, but why is it important to distinguish sensible from latent fluxes here?

Our intention is to show important contributing factors to mean $F_h$, the sum of surface sensible and latent heat fluxes. The surface sensible and latent fluxes share surface wind speed, but thermodynamic variables ($\Delta q$ and $\Delta T$) are different. So, it is interesting to examine if the thermodynamic conditions also differ and whether it is mainly due to temperature or moisture. For example, convective downdrafts transport colder and moister air into the boundary layer, which can lead to substantial influences of thermodynamics on surface fluxes. It is not our case, but it may be if one uses a slab ocean model (Tompkins and Semie, 2021), as the reviewer mentioned. Therefore, it is worth exploring and demonstrating the different thermodynamic influences separately. In the revised manuscript, we have clarified why we distinguish surface latent and sensible heat fluxes in the context of Sect. 5.1.1.

L352-353: "Here we begin with partitioning $F_h$ into surface sensible and latent heat fluxes to examine the importance of thermodynamic variables, i.e., $\Delta q$ and $\Delta T$ as well as $\overline{U}_h$ for mean $F_h$."

Line 633-634:  If radiative cooling is shut off, there can be no latent heating that, over the whole domain, must balance the cooling. The system would shut down.

What we mean is a change or perturbation of radiative cooling. It does not mean that there is no radiative cooling at all. We have clarified this in the revised manuscript.

L714-715: "The model configuration changes radiative cooling and dry stability in all latitudes, but these changes compensate each other, having a very small net effect on convective mass flux."

---

## Author Response (AR2)

Response to comments of Referee #3

We thank the reviewer for the thoughtful comments on our manuscript. We have carefully considered each of the comments. The reviewer's comments are repeated in normal font and our responses are followed in blue.

The revised manuscript has mostly addressed the concerns from the previous reviewers. The main issue in the last review round was the resolution (13km) being too coarse to resolve convection. In the revised manuscript, the authors have included two higher-res experiments (5km, P5 and E5). It is nice that in the conclusion section, the authors show a schematic and summarise the importance of different processes in each configure. Before the final decision, I still have some concerns (mostly minor).

For the schematic plot, discussions about the first two panels (dependence on convective treatment at 13km and dependence on resolution with the same convective treatment) are quite clear, but the discussion about the dependence on convective treatment at 5km is bit too short and general. The authors mention that 5km is more complex than 13 km. Why? I would like to see more about this. For instance, what is consistent and what is different?

Thank you for pointing this out. We have elaborated the description of the dependence on convective treatment at 5km in the conclusions.

L702-712: "The sensitivity of rainfall to convective treatment at 5km exhibits some differences from that at 13km. As for the other cases, explicit convection produces more rainfall than parameterized convection. The difference in rainfall is associated primarily with differences in convective updraft mass flux related to BLQE, while precipitation efficiency remains largely unaffected by convective treatment. At 5km, however, the differences in surface enthalpy fluxes are not due to differences in surface wind speed, which is relatively similar in the two runs, although the Hadley circulation is stronger in E5 than in P5. Instead, a larger moisture contrast between the ocean surface and the air in E5 enhances surface enthalpy fluxes and convective mass flux. Furthermore, the effects of stronger convective mass flux on rainfall is partially offset by relatively low column averaged humidity, of which the differences in moisture in the lower troposphere contribute the most. Thus, BLQE is still key to understand the dependence on convective treatment at 5km, yet the balance is achieved by thermodynamics within and above the BL, while dynamic fields are less involved."

Another concern from the previous reviewer is that this is a diagnostic framework which does not help us understand causality. The authors have clarified in the reply that they cannot disentangle what drives what and addressing 'links' is not their main focus. This is a fair reply. However, in the manuscript, the authors should be careful when describing the relationships. Because in the end, there may be explanation for causality in both directions.

Below is an example:
-Line/section 700, last two lines, 'the vertical contrast in MSE becomes larger with increasing resolution….' I am a bit confused by this sentence. How would large MSE difference broaden ITCZ? The authors seem to argue drier free-troposphere would increase more import of low MSE air into the boundary layer, which then would increase surface fluxes. However, based on EQ 1 and figure 4, wouldn't that be the opposite? More low MSE air into the boundary layer would reduce MSE in the PBL. This would narrow ITCZ.

We admit that this is a little misleading and that we somewhat overemphasized the "broadening" of the ITCZ. As described in Sect. 3.1, the broader ITCZ, which especially extends further into the northern hemisphere, seems to be associated with an asymmetry in the initialization already. E5 and P5 are both initialized with the output from P13, which has the MJO-like disturbance for its last simulation days. This disturbance is accompanied by enhanced rainfall in the northern hemisphere, creating an asymmetry in rainfall pattern. This asymmetry in P13 is carried on for P5 somehow more clearly than for E5, in which we still see a hint of an asymmetry, e.g., slightly increased rainfall in 7-10°N. In the revised manuscript we have played down it and have focused on the reduced rainfall in the ITCZ due to horizontal resolution.

L12-13:: "Increasing horizontal resolution substantially reduces the rainfall maximum in the ITCZ, while the strength of the Hadley circulation changes only marginally."

L617: "Decreased $M_u$ with increasing resolution is associated with the combined effects of increased $h_b$-$h_m$ and suppressed $F_h$, which is shaped by $\overline{U}_h$ through the large-scale circulation."

L679: "These changes are more pronounced at 13km than at 5km, which shows reduced rainfall in the ITCZ."

L696-700: "Additionally, the vertical contrast in moist static energy in the ITCZ becomes larger with increasing resolution. The combination of suppressed surface enthalpy fluxes and increased vertical contrast in moist static energy are associated with reduced convective mass flux, while precipitation efficiency changes little due to increasing resolution. This underlines the importance of BLQE to understand rainfall difference due to changing resolution."

I am surprised that precipitation efficiency does not change much across these different setups, given the vast differences in the convective treatment. I wonder if the authors have some ideas about that.

We were surprised about this as well but don't have an obvious explanation for it. We planned to look into this point from a different angle by calculating condensation rate tendency from the microphysics scheme, but it would be inconsistent to compare that between parameterized and explicit convection. Our ITCZ diagnostic can only tell precipitation efficiency averaged over time and longitude. The averaging collects different precipitation efficiencies from individual precipitating clouds. Certainly, these clouds would have a wide range of values in precipitation efficiency, but the mean precipitation efficiency is quite similar between the simulations. We

can speculate that the convection scheme in the model is somehow tuned to produce an precipitation efficiency close the explicit treatment. We leave a deeper analysis for future research.

For the analyses in this study across different resolutions, were the data with different resolutions regridded to the same grid?
Yes, we remapped model grids to 0.2° lat-lon. This is described in L197.

Section 2 'Aqauchannel' —> Aquachannel
We have corrected the typo in the revised version.